



# Direct Inversion of Circulation and Mixing from Tracer Measurements: I. Method

Thomas von Clarmann[1] and Udo Grabowski[1]

[1]Karlsruhe Institute of Technology, Institute of Meteorology and Climate Research,
Karlsruhe, Germany

*Correspondence to:* T. von Clarmann (thomas.clarmann@kit.edu)

**Abstract.** From a series of zonal mean global stratospheric tracer measurements sampled in altitude versus latitude, circulation and mixing patterns are inferred by the inverse solution of the continuity equation. As a first step, the continuity equation is written as a tendency equation, which is numerically integrated over time to predict a later atmospheric state, i.e. mixing ratio and air density. The integration is formally performed by multiplication of the initially measured atmospheric state vector by a

linear prediction operator. Further, the derivative of the predicted atmospheric state with respect to the wind vector components and mixing coefficients is used to find the most likely wind vector components and mixing coefficients which minimize the residual between the predicted atmospheric state and the later measurement of the atmospheric state. Unless multiple tracers are used, this inversion problem is under-determined, and dispersive behaviour of the prediction further destabilizes the inversion. Both these problems are fought by regularization. For this purpose, a first order smoothness constraint has been chosen.

The usefulness of this method is demonstrated by application to various tracer measurements recorded with the Michelson Interferometer for Passive Atmospheric Sounding (MIPAS). This method aims at a diagnosis of the Brewer-Dobson circulation without involving the concept of the mean age of stratospheric air, and related problems like the stratospheric tape recorder, or intrusions of mesospheric air into the stratosphere.

## 1   Introduction

In the context of climate change, possible changes of the intensity of the Brewer-Dobson circulation are under debate. Climate models predict an intensification of the Brewer-Dobson circulation (Butchart et al., 2006). Engel et al. (2009), however, found a weakly significant slow increase of the mean age of stratospheric air. The latter is defined as the mean time lag between the date of the transition of tropospheric air into the stratosphere and the date when the mixing ratio of a monotonically growing tracer was measured in the air volume under investigation, and its increase hints at a deceleration of the Brewer-Dobson

circulation. These measurements have been challenged as not representative (Garcia et al., 2011), and global mean age of air measurements by Stiller et al. (2012) suggest that the true picture is not that one-dimensional. Instead, stratospheric age trends vary with altitude and with latitude. Determination of the age of air and its use as diagnostic of the intensity of the Brewer-Dobson circulation, however, has its own limitations: First, due to mixing processes, the age of a stratospheric air volume is not unique but characterized by an age spectrum, on which some *ad hoc* assumptions have to be made (Waugh and Hall, 2002).



Second, the most suited age tracer, $SF_6$, which has significant and monotonic growth rates in the troposphere, is not fully inert: It has a mesospheric sink (Hall and Waugh, 1998; Reddmann et al., 2001) and introduces some age uncertainty when mesospheric air subsides into the stratosphere in the polar winter vortex (Stiller et al., 2008). Third, the determination of the mean age relies upon a reference airmass where the age, by definition, is zero. When the age of air concept was introduced, the reference was simply the troposphere, which is well mixed and thus avoids any related complication (Solomon, 1990; Schmidt and Khedim, 1991). Since the age of air has become a model diagnostic, the modelling community has established the upper edge of the tropical tropopause layer as a reference (Hall and Plumb, 1994), which makes a difference due to the slow ascent of air through the tropical tropopause layer (Fueglistaler et al., 2009). For model validation, however, this redefined age of air is of limited use, because no long measured time series of tracer mixing ratios are available there.

Facing these difficulties, it is desirable to infer the atmospheric circulation directly from tracer measurements, without going back to the age of air concept. Multiple approaches have been developed to infer windfields from measured atmospheric state variables. Sequential data assimilation, and in its optimal form, the extended Kalman filter approach (e.g. Ghil and Malanotte-Rizzoli, 1991; Ghil, 1997), calculates the optimal average of the forecasted meteorological variables for the time of the observation and the observed meteorological variables themselves and uses this average to initialize the next forecast step. The wind field is calculated by a dynamical model. This method involves the generalized inversion of the observation operator where the forecast is used as a constraint. In contrast, so-called[1] variational data assimilation minimizes the residual between the forecasted and the measured atmospheric state variables by optimally adjusting the initialization of the forecast via inversion of an adjoint forecast model, constrained by some background state (Thompson, 1969). Both approaches rely on dynamical models[2] and are suited to infer the most probable atmospheric state variables rather than the windfield, which is a by-product of the assimilation. The windfield or atmospheric circulation can also be inferred directly by kinematic methods from tracer measurements. Such methods rely solely on the continuity equation, do not involve a dynamic model, and thus do not depend on any *ad hoc* parametrisation of effects which are either not resolved by the discrete model, computationally too expensive for explicit modeling, or simply not well understood. While this work is targeted primarily at an assessment of the Brewer-Dobson circulation, its applicability is much wider and includes stratospheric-tropospheric exchange, the mesospheric overturning circulation and others. Early approaches to infer the circulation from tracer measurements include Holton and Choi (1988) as well as Salby and Juckes (1994) who used approaches which share several ideas with ours. Direct inversion of wind speeds from tracer measurements in a volcano plume has, e.g., been suggested by Krueger et al. (2013), however without consideration of mixing. The continuity equation including diffusion terms has been exploited by Wofsy et al. (1994) for assessment of diffusion of stratospheric aircraft exhaust.

In this paper we present a method to infer two-dimensional (latitude/altitude) circulation and mixing coefficients from subsequent measurements of inert tracers. The application of this method, i.e., the inference of the Brewer-Dobson circulation from global $SF_6$ distributions (Stiller et al., 2008, 2012) measured with the Michelson Interferometer for Passive Atmospheric Sounding (MIPAS), is presented in a companion paper. In order to avoid that the reader does not see the forest for the trees,

---

[1]The term 'so-called' is used here, because it is challenged that this method is really variational in the context of discrete variables (Wunsch, 1996)[p368].

[2]This statement refers to meteorological data assimilation. Chemical data assimilation uses chemistry transport models.





we give a short overview of our method in Section 2. The prediction of pressure and tracer mixing ratio fields on the basis of the continuity equation and related error estimation is described in Section 3. The estimation of circulation and mixing coefficients by inversion of the continuity equation is presented in Section 4. In Section 5, the applicability of the method and the need of further refinements is critically discussed. The benefits of the method are discussed in Section 6. The paper

concludes with recommendations how these results should be used and with an outlook on future work (Section 7). Changes of the Brewer-Dobson circulation during 2002-2012, i.e. the MIPAS mission, are currently investigated by means of this method and will be published in a subsequent paper.

## 2   General concept

Knowing the initial state of the atmosphere in terms of mixing ratio and air density distributions, wind speed and mixing

coefficients at each gridpoint, a future atmospheric state can be predicted with respect to the distribution of any inert tracer. This procedure we call the forward problem. If no ideal tracers are available, source terms of related species have to be included in the forward model. The goal of this work is to invert the forward model in order to infer the circulation and mixing coefficients from tracer measurements by minimization of the residual between the predicted and measured atmospheric state. This approach is complementary to free running climate models because it makes no assumptions about atmospheric dynamics

except the validity of the continuity equation. It is further considered more robust than age-of-air analysis (Stiller et al., 2012) because it does not depend on a reference point where the age is assumed zero, nor does it require knowledge on the history of an air parcel.

## 3   The forward problem

The forward model reads the measured atmospheric state at time $t$ and predicts the atmospheric state (number density of air, $c$,

and volume mixing ratios, $vmr$) at time $t + \Delta t$ for given wind vectors and mixing coefficients representing the time interval $[t; t + \Delta t]$ by solving the continuity equation. The continuity equation allows to calculate the local tendencies of the number densities and volume mixing ratios. These local tendencies $\frac{\partial \rho}{\partial t}$ and $\frac{\partial vmr}{\partial t}$ are then integrated over time to give the new number densities and mixing ratios.

### 3.1   The continuity equation

The local change of number density $\rho$ of air is in spherical coordinates (for all auxiliary calculations, see supplement):

$$\frac{\partial \rho}{\partial t} = -\frac{1}{r}\frac{\partial \rho v}{\partial \phi} + \frac{\rho v}{r}\tan(\phi) - \frac{\partial \rho w}{\partial z} - \frac{2\rho w}{r} - \frac{1}{r\cos(\phi)}\frac{\partial \rho u}{\partial \lambda} \tag{1}$$

where





$$
\begin{array}{rcl}
t & = & \text{time} \\
\lambda & = & \text{longitude} \\
\phi & = & \text{latitude} \\
z & = & \text{altitude above surface} \\
r & = & r_E + z \\
r_E & = & \text{radius of Earth} \\
u & = & (r_E + z)\cos\phi\, \mathrm{d}\lambda/\mathrm{d}t \\
v & = & (r_E + z)\mathrm{d}\phi/\mathrm{d}t \\
w & = & \mathrm{d}z/\mathrm{d}t
\end{array}
$$

Here the shallowness approximation (Hinkelmann (1951); Phillips (1966), quoted after Kasahara (1977)), which is, often implicitly, used in the usual textbooks on atmospheric sciences (e.g. Brasseur and Solomon, 2005, their Eq. 3.46a), is intentionally not used for reasons which will become clear in Section 3.2.

5   The local change of the volume mixing ratio of gas $g$ can be calculated from known velocities and mixing coefficients as well as source/sink terms as

$$
\frac{\partial vmr_g}{\partial t} = \frac{S_g}{\rho} - \frac{u}{r\cos\phi}\frac{\partial vmr_g}{\partial \lambda} - \frac{v}{r}\frac{\partial vmr_g}{\partial \phi} - w\frac{\partial vmr_g}{\partial z} + \frac{1}{r^2}\frac{\partial}{\partial \lambda}\left[\frac{K_\lambda}{\cos^2\phi}\frac{\partial vmr_g}{\partial \lambda}\right] \\
+ \frac{1}{r^2\cos\phi}\frac{\partial}{\partial \phi}\left[K_\phi\cos\phi\frac{\partial vmr_g}{\partial \phi}\right] + \frac{1}{r^2}\frac{\partial}{\partial z}\left[r^2 K_z\frac{\partial vmr_g}{\partial z}\right]
\tag{2}
$$

where

$$
\begin{array}{rcl}
vmr_g & = & \text{volume mixing ratio of species } g \\
K_\lambda & = & \text{zonal diffusion coefficient} \\
K_\phi & = & \text{meridional diffusion coefficient} \\
K_z & = & \text{vertical diffusion coefficient} \\
S_g & = & \text{the production/loss rate of species } g \text{ in} \\
& & \text{terms of number density over time}
\end{array}
$$

(Brasseur and Solomon (e.g. 2005, Eq. 3.46b) and Jones et al. (2007)).

Since we are only interested in a two-dimensional representation of the atmosphere in altitude and latitude coordinates, zonal advection and mixing terms are ignored in Eqs. (1–2). In this two-dimensional representation, all atmospheric state variables represent zonal mean values, and the diffusion coefficient $K_\phi$ does not only describe physical diffusion but also eddy diffusion
15   components arising from the symmetric part of the two-dimensional eddy tensor. Accordingly, the velocities are no longer windspeed only but include also an eddy component arising from the antisymmetric part of the eddy tensor (Ko et al., 1985, and references therein) and thus are effective transport velocities. Therefore, the transition from the 3D to the 2D system and involved absorption of eddy flux terms in mixing coefficients and velocities implies a re-interpretation of the relevant quantities.




Details of the transition from the three-dimesnional to the two-dimensional problem are discussed in Appendix A. The local change of number density $\rho$ of air in a two-dimensional atmosphere thus is

$$\frac{\partial \rho}{\partial t} = -\frac{1}{r}\frac{\partial \rho v}{\partial \phi} + \frac{\rho v}{r}\tan(\phi) - \frac{\partial \rho w}{\partial z} - \frac{2\rho w}{r} \tag{3}$$

and the local change of $vmr_g$ is calculated as

$$\frac{\partial vmr_g}{\partial t} = \frac{S_g}{\rho} - \frac{v}{r}\frac{\partial vmr_g}{\partial \phi} - w\frac{\partial vmr_g}{\partial z} + \frac{1}{r^2\cos\phi}\frac{\partial}{\partial \phi}\left[K_\phi\cos\phi\frac{\partial vmr_g}{\partial \phi}\right] + \frac{1}{r^2}\frac{\partial}{\partial z}\left[r^2 K_z\frac{\partial vmr_g}{\partial z}\right] \tag{4}$$

In order to comply with the continuity equation, zonal averages of $vmr_g$ have to be calculated number-density-weighted.

## 3.2 Integration of tendencies

Integration of Eqs (3–4) is performed numerically for timesteps of $\Delta t_p$. For practical reasons, processes (advection, diffusion, sinks) are splitted, i.e. the tendencies due to these three classes of processes are integrated independently. The timesteps $\Delta t_p$ used for the integration are chosen smaller than the time difference $\Delta t$ between two measurements, in order not to clash with the Courant limit (Courant et al., 1952). In the following we call $\Delta t_p$ 'micro time increment' and the latter 'macro time increment'. The atmospheric state after a macro time increment is predicted by successive prediction over the micro time increment. In the following, index $i$ designates time $t$, $i+1$ designates the time $t + \Delta t_p$, etc, and $I$ designates the time after the final micro time increment, i.e. the next macro time increment.

For the discrete integration of the advection part of the tendencies the MacCormack (1969) method is used in a generalized multidimensional version similar to the one described by (Perrin and Hu, 2006). This is a predictor-corrector method. For a general state variable $c(t,x,y) = c_i(x,y)$ at location $(x,y)$, and time $t$ with $e(c)$ and $f(c)$ being functions of $c$, an equation of the form

$$\frac{\partial c}{\partial t} + \frac{\partial e(c)}{\partial x} + \frac{\partial f(c)}{\partial y} = 0 \tag{5}$$

is solved by preliminary predictions of the state variable as a first step: $x$ is

$$c^*_{i+1}(x,y) = c_i(x,y) - \frac{\Delta t_p}{\Delta x}\left(e_i(x+\Delta x,y) - e_i(x,y)\right) - \frac{\Delta t_p}{\Delta y}\left(f_i(x,y+\Delta y) - f_i(x,y)\right). \tag{6}$$

These are then used in a subsequent correction step which gives the final prediction:

$$c_{i+1}(x,y) = \tag{7}$$

$$\frac{1}{2}\left[c_i(x,y) + c^*_{i+1}(x,y) - \frac{\Delta t_p}{\Delta x}\left(e(c^*_{i+1},x,y) - e(c^*_{i+1},x-\Delta x,y)\right) - \frac{\Delta t_p}{\Delta y}\left(f(c^*_{i+1},x,y) - f(c^*_{i+1},x,y-\Delta y)\right)\right]$$

Application to the continuity equation in spherical coordinates requires reformulation of Eq. (3) (c.f., e.g., Chang and St.-Maurice, 1991):

$$\frac{\partial r^2\rho\cos(\phi)}{\partial t} = -\frac{\partial r\rho\, v\,\cos(\phi)}{\partial \phi} - \frac{\partial r^2\rho\, w\,\cos(\phi)}{\partial z} \tag{8}$$



The predictor of $r^2\rho\cos(\phi)$ is then calculated as

$$[r^2\rho_{i+1}(\phi,z))\cos(\phi)]^* = \quad r^2\rho_i(\phi,z)\cos(\phi) - \frac{\Delta t_p}{\Delta\phi}\left(r\rho_i(\phi+\Delta\phi,z)\cos(\phi+\Delta\phi)v(\phi+\Delta\phi,z) - r\rho_i(\phi,z)\cos(\phi)v(\phi,z)\right) \quad (9)$$
$$-\frac{\Delta t_p}{\Delta z}\left((r+\Delta z)^2\rho_i(\phi,z+\Delta z)\,w_{\phi,z+\Delta z}\,\cos(\phi) - r^2\rho_i(\phi,z)\,w_{\phi,z}\,\cos(\phi)\right)$$

and the corrected prediction for $\rho$ then gives

$$\rho_{i+1}(\phi,z) = \frac{1}{2r^2\cos(\phi)} \times \left[\rho_i(\phi,z)r^2\cos(\phi) + \left[\rho_{i+1}(\phi,z)r^2\cos(\phi)\right]^* - \frac{\Delta t_p}{\Delta\phi}\left[\left[\rho_{i+1}(\phi,z)rv(\phi,z)\cos(\phi)\right]^*\right.\right. \quad (10)$$
$$\left. - \left[\rho_{i+1}(\phi-\Delta\phi,z)rv(\phi-\Delta\phi,z)\cos(\phi-\Delta\phi)\right]^*\right] - \frac{\Delta t_p}{\Delta z}\left[\left[\rho_{i+1}(\phi,z)r^2w(\phi,z)\cos(\phi)\right]^*\right.$$
$$\left.\left. - \left[\rho_{i+1}(\phi,z-\Delta z)(r-\Delta z)^2w(\phi,z-\Delta z)\cos(\phi)\right]^*\right]\right]$$

For the local change of mixing ratio, operator splitting is performed. The horizontal and vertical advective parts of the continuity equation for mixing ratios in two dimensions are transformed into the following Mac-Cormack-integrable forms:

$$\left[\frac{\partial\frac{r\,vmr}{v}}{\partial t}\right]_{adv.horiz} = -\frac{\partial vmr_g}{\partial\phi} \quad (11)$$

and

$$\left[\frac{\partial\frac{vmr_g}{w}}{\partial t}\right]_{adv.vert} = \frac{\partial vmr_g}{\partial z}, \quad (12)$$

respectively.

For the diffusive component we use simple Eulerian integration:

$$[vmr_{g;i+1}(\phi,z) - vmr_{g;i}\phi,z]_{\text{diff}} = \frac{\Delta t_p}{2r^2(\Delta\phi)^2\cos(\phi)} \cdot \quad (13)$$
$$\left[(K_\phi(\phi,z)+K_\phi(\phi+\Delta\phi))\cos(\phi+\frac{\Delta\phi}{2})(vmr_{g;i}(\phi+\Delta\phi,z)-vmr_{g;i}(\phi,z)) - (K_\phi(\phi,z)+K_\phi(\phi-\Delta\phi))\cos(\phi-\frac{\Delta\phi}{2}) \cdot\right.$$
$$\left.(vmr_{g;i}(\phi,z)-vmr_{g;i}(\phi-\Delta\phi,z))\right] + \frac{\Delta t_p}{2r^2(\Delta z)^2}\left[(r+\frac{\Delta z}{2})^2 \cdot (K_z(\phi,z)+K_z(\phi,z+\Delta z)) \cdot\right.$$
$$\left.(vmr_{g;i}(\phi,z+\Delta z)-vmr_{g;i}(\phi,z)) - (r-\frac{\Delta z}{2})^2 \cdot (K_z(\phi,z)+K_z(\phi,z-\Delta z)) \cdot (vmr_{g;i}(\phi,z)-vmr_{g;i}(\phi,z-\Delta z))\right]$$

Sinks of the species considered here are treated as unimolecular processes (c.f., e.g. Brasseur and Solomon, 2005, their Eq. 2.27d) and integrated as

$$\rho_{g;i+1} = \rho_{g;i}e^{-k_g\Delta t_p} \quad (14)$$

where $k_g$ is the sink strength of the gas $g$.

The abundance of gas $g$ after time-step $\Delta t_p$ is then simply the sum of the increments due to horizontal and vertical advection, diffusion, and chemical losses.





Admittedly, there exist more elaborate advection schemes than the one used here. However, the need to provide the Jacobians needed in Sections 3.3–4 justifies a reasonable amount of simplicity. Further, numerical errors cannot easily accumulate, because after each timestep $\Delta t$, the system is re-initialized with measured data.

Since we do not have a closed system but have mass exchange and mixing with the atmosphere below the lowermost model altitude and above the uppermost altitude, the atmospheric state is not predicted for the lowermost and uppermost altitudes. Prediction is only possible from the second to one below the uppermost altitude. This restricted altitude range we henceforth call 'nominal altitude range'. Instead, the atmospheric state of the uppermost and lowermost altitude is estimated by linear interpolation of measured values at times $t$ and $t + \Delta t$ and used as boundary condition for prediction within the nominal altitude range. Alternatively, derivatives at the border can be approximated by asymmetric difference quotients.

We use the following convention: Atmospheric state variables are sampled on a regular latitude-altitude grid. For some gridpoints, no valid measurements may be available but we assume that for each state variable we have a contiguous subset of this grid with valid measurements. For state variable $g$, we have a total of $J_g$ valid measurements within the 'nominal altitude range', each denoted by index $j$. A state variable in this context is either air density $\boldsymbol{\rho}$ ($g = 0$) or the mixing ratio of one species $\boldsymbol{vmr}_g$. The nominal altitude range at latitude $\phi$ is the altitude range where, for each gridpoint, a valid measurement is available at the gridpoint itself, and for its northern, southern, upper, lower and diagonal neighbours. The use of asymmetric difference quotients can be emulated by generating artificial border values by extrapolation which guarantee that each gridpoint within the nominal altitude and latitude range has all required northern, southern, upper, lower and diagonal neighbours. The availability of neighbour-values is necessary to allow the calculation of numerical derivatives of the state variable with respect to latitude and altitude. Further, we have $K_g$ border elements of each quantity $g$, each denoted by index $k$. And finally, for each state variable $g$, we have a total of $L_g = J_g + K_g$ gridpoints, with indices $l$.

### 3.3 Integration in operator notation

For further steps (error propagation and the solution of the inverse problem) it is convenient to rewrite the prediction of air density and mixing ratios in matrix notation. For this purpose, we differentiate the predicted air densities (Eq. 10) and mixing ratios (Eqs. 11–13) with respect to air density and mixing ratios of the gases under assessment at all relevant locations. The sensitivities of the densities of the first predictive step with respect to the initial densities at the same latitude and altitude are

$$\frac{\partial \rho_{i+1}(\phi, z)}{\partial \rho_i(\phi, z)} = \frac{1}{2}\left[2 - \frac{\Delta t_p}{\Delta \phi}\left[\frac{v(\phi, z)}{r}\left(\frac{\Delta t_p}{\Delta \phi} \cdot \frac{v(\phi, z)}{r} + \frac{\Delta t_p}{\Delta z}w(\phi, z)\right) + \frac{\Delta t_p}{\Delta \phi}\frac{v(\phi - \Delta \phi, z)v(\phi, z)}{r^2}\right]\right.$$
$$\left. - \frac{\Delta t_p}{\Delta z}\left[w(\phi, z)\left(\frac{\Delta t_p}{\Delta \phi} \cdot \frac{v(\phi, z)}{r} + \frac{\Delta t_p}{\Delta z}w(\phi, z)\right) + \frac{\Delta t_p}{\Delta z}w(\phi, z - \Delta z)w(\phi, z)\right]\right] \tag{15}$$

We further differenciate predicted air densities with respect to air densities at the adjacent southern latitude but the same altitude.

$$\frac{\partial \rho_{i+1}(\phi, z)}{\partial \rho_i(\phi - \Delta \phi, z)} = \frac{1}{2}\left[\frac{\Delta t_p}{\Delta \phi} \cdot \frac{v(\phi - \Delta \phi, z)}{r} \cdot \frac{\cos(\phi - \Delta \phi)}{\cos(\phi)}\left(1 + \frac{\Delta t_p}{\Delta \phi} \cdot \frac{v(\phi - \Delta \phi, z)}{r} + \frac{\Delta t_p}{\Delta z}w(\phi - \Delta \phi, z)\right)\right] \tag{16}$$





The derivative of the predicted air densities with respect to air densities at the adjacent northern latitude but the same altitude is

$$\frac{\partial \rho_{i+1}(\phi,z)}{\partial \rho_i(\phi+\Delta\phi,z)} = \frac{1}{2}\left[\frac{\Delta t_p}{\Delta\phi}\cdot\frac{v(\phi+\Delta\phi,z)}{r}\cdot\frac{\cos(\phi+\Delta\phi)}{\cos(\phi)}\left(-1+\frac{\Delta t_p}{\Delta\phi}\cdot\frac{v(\phi,z)}{r}+\frac{\Delta t_p}{\Delta z}w(\phi,z)\right)\right].$$ (17)

As a next step we differenciate predicted air densities with respect to the initial air densities at the next higher altitude but the same latitude.

$$\frac{\partial \rho_{i+1}(\phi,z)}{\partial \rho_i(\phi,z+\Delta z)} = \frac{1}{2}\left[\frac{\Delta t_p}{\Delta z}\cdot\frac{(r+\Delta z)^2}{r^2}w(\phi,z+\Delta z)\cdot\left(-1+\frac{\Delta t_p}{\Delta\phi}\cdot\frac{v(\phi,z)}{r}+\frac{\Delta t}{\Delta z}w(\phi,z)\right)\right]$$ (18)

The derivative of the predicted air densities with respect to the initial air densities at the next lower altitude but the same latitude is

$$\frac{\partial \rho_{i+1}(\phi,z)}{\partial \rho_i(\phi,z-\Delta z)} = \frac{1}{2}\left[\frac{\Delta t_p}{\Delta z}w(\phi,z-\Delta z)\frac{(r-\Delta z)^2}{r^2}\cdot\left(1+\frac{\Delta t_p}{\Delta\phi}\frac{v(\phi,z-\Delta z)}{r-\Delta z}+\frac{\Delta t_p}{\Delta z}w(\phi,z-\Delta z)\right)\right].$$ (19)

Finally we differentiate the predicted air densities with respect to the initial air densities at the adjacent southern latitude and higher altitude

$$\frac{\partial \rho_{i+1}(\phi,z)}{\partial \rho_i(\phi-\Delta\phi,z+\Delta z)} = -\frac{1}{2}\left[\frac{v(\phi-\Delta\phi,z)}{r}\cdot\frac{\Delta t_p}{\Delta\phi}\cdot\frac{\Delta t_p}{\Delta z}\cdot\frac{(r+\Delta z)^2}{r^2}\cdot\frac{\cos(\phi-\Delta\phi)}{\cos(\phi)}w(\phi-\Delta\phi,z+\Delta z)\right]$$ (20)

and vice versa

$$\frac{\partial \rho_{i+1}(\phi,z)}{\partial \rho_i(\phi+\Delta\phi,z-\Delta z)} = -\frac{1}{2}\left[w(\phi,z-\Delta z)\frac{\Delta t_p}{\Delta z}\cdot\frac{\Delta t_p}{\Delta\phi}\cdot\frac{(r-\Delta z)^2}{r^2}\cdot\frac{\cos(\phi+\Delta\phi)}{\cos(\phi)}\cdot\frac{v(\phi+\Delta\phi,z-\Delta z)}{r-\Delta z}\right]$$ (21)

where $i$ is the index of the time increment, and where $\phi\pm\Delta\phi$ and $z\pm\Delta z$ refer to the adjacent model gridpoints in latitude and altitude, respectively.

For mixing ratios, the respective derivatives are:

$$\frac{\partial vmr_{i+1}(\phi,z)}{\partial vmr_i(\phi,z)} = 1 - \left(\frac{\Delta t_p}{\Delta\phi}\right)^2\cdot\frac{v(\phi,z)}{r^2}\cdot\frac{1}{2}\left[v(\phi,z)+v(\phi-\Delta\phi,z)\right]$$ (22)

$$-\left(\frac{\Delta t_p}{\Delta z}\right)^2\cdot w(\phi,z)\cdot\frac{1}{2}\left[w(\phi,z)+w(\phi,z-\Delta z)\right]-\frac{\Delta t_p}{2r^2(\Delta\phi)^2\cos(\phi)}\left[\left(K_\phi(\phi,z)+K_\phi(\phi+\Delta\phi,z)\right)\cdot\cos\left(\phi+\frac{\Delta\phi}{2}\right)\right.$$

$$+\left(K_\phi(\phi,z)+K_\phi(\phi-\Delta\phi,z)\right)\cos\left(\phi-\frac{\Delta\phi}{2}\right)\right]-\frac{\Delta t_p}{2r^2(\Delta z)^2}\left[\left(r+\frac{\Delta z}{2}\right)^2\left(K_z(\phi,z)+K_z(\phi,z+\Delta z)\right)\right.$$

$$+\left(r-\frac{\Delta z}{2}\right)^2\left(K_z(\phi,z)+K_z(\phi,z-\Delta z)\right)\right] - Loss(month,\phi,z)\Delta t_p;$$

$$\frac{\partial vmr_{i+1}(\phi,z)}{\partial vmr_i(\phi+\Delta\phi,z)} =$$ (23)

$$-\frac{\Delta t_p}{\Delta\phi}\cdot\frac{v(\phi,z)}{2r}\left(1-\frac{\Delta t_p}{\Delta\phi}\cdot\frac{v(\phi,z)}{r}\right)+\frac{\Delta t_p}{2r^2(\Delta\phi)^2\cos(\phi)}\cdot\left(K_\phi(\phi,z)+K_\phi(\phi+\Delta\phi,z)\right)\cos\left(\phi+\frac{\Delta\phi}{2}\right);$$





$$\frac{\partial vmr_{i+1}(\phi,z)}{\partial vmr_i(\phi-\Delta\phi,z)} = \tag{24}$$
$$\frac{v(\phi,z)}{2r} \cdot \frac{\Delta t_p}{\Delta\phi}\left(1 + \frac{\Delta t_p}{\Delta\phi}\cdot\frac{v(\phi-\Delta\phi,z)}{r}\right) + \frac{\Delta t_p}{2r^2(\Delta\phi)^2\cos(\phi)}\cdot\left(K_\phi(\phi,z)+K_\phi(\phi-\Delta\phi,z)\right)\cos(\phi-\frac{\Delta\phi}{2});$$

$$\frac{\partial vmr_{i+1}(\phi,z)}{\partial vmr_i(\phi,z+\Delta z)} = \tag{25}$$
$$-\frac{1}{2}\cdot w(\phi,z)\cdot\frac{\Delta t_p}{\Delta z}\left(1 - \frac{\Delta t_p}{\Delta z}w(\phi,z)\right) + \frac{\Delta t_p}{2r^2(\Delta z)^2}(r+\frac{\Delta z}{2})^2\left(K_z(\phi,z)+K_z(\phi,z+\Delta z)\right);$$

$$\frac{\partial vmr_{i+1}(\phi,z)}{\partial vmr_i(\phi,z-\Delta z)} = \tag{26}$$
$$\frac{\Delta t_p}{\Delta z}\cdot\frac{1}{2}\cdot w(\phi,z)\left(1 + w(\phi,z-\Delta z)\frac{\Delta t_p}{\Delta z}\right) + \frac{\Delta t_p}{2r^2(\Delta z)^2}(r-\frac{\Delta z}{2})^2\left(K_z(\phi,z)+K_z(\phi,z-\Delta z)\right),$$

10 where $Loss(month,\phi,z)$ is the relative loss rate in the respective month at latitude $\phi$ and altitude $z$. These derivatives are simplifications in a sense that they do not consider the full chemical Jacobian but assume instead that the source strength depends on no other concentration than the actual concentration of the same species. For the typical long-lived so-called tropospheric source gases considered here, like $SF_6$ or CFCs, this assumption is appropriate. Pretabulated loss rates are used which have been calculated by locally integrating loss rates over an entire month at a time resolution adequate to resolve

15 the diurnal cycle. From the monthly losses, the $Loss(month,\phi,z)$ values, which are the contribution of losses to the partial derivatives of the local mixing ratios with respect to the initial local mixing ratios, are calculated as the secant of the local decay curve.

With these expressions, the prediction of air density and volume mixing ratio can be rewritten in matrix notation for a single micro time increment:

$$\boldsymbol{\rho}_{i+1} = \begin{pmatrix} \rho_{l=1} \\ \vdots \\ \rho_{L_0} \end{pmatrix}_{i+1} = \mathbf{D}_{\rho;i}\boldsymbol{\rho}_i = \begin{pmatrix} \mathbf{I}_K & \mathbf{0} & \mathbf{0} \\ \mathbf{W}_i & \mathbf{0} \\ \mathbf{0} & \mathbf{D}_{\rho,\text{nom}} \end{pmatrix}\begin{pmatrix} \boldsymbol{\rho}_{I;k=1,K_0} \\ \boldsymbol{\rho}_{i;k=1,K_0} \\ \boldsymbol{\rho}_{i;j=K_0+1,L_0} \end{pmatrix} \tag{27}$$





where

$\mathbf{D}_{\rho;i}$   is the $L_0 \times L_0$ Jacobian matrix of air density for time increment $i$, i.e. the sensitivities of the prediction with respect to the initial state, $\frac{\partial c_{i+1,m}}{\partial c_{i,n}}$, here $m$ and $n$ run over the model gridpoints

$\mathbf{I}_K$   is $K_0 \times K_0$ identity;

$\mathbf{0}$   are zero submatrices of the required dimensions;

$\mathbf{W}_i$   is a $K_0 \times 2K_0$-dimensional interpolation matrix;

$\mathbf{D}_{\rho;\text{nom}}$   is an $J_0 \times L_0$ Jacobian containing the partial derivatives $\partial \rho_{i+1;j}/\partial \rho_{i;l}$, applied to the nominal altitude range;

$\boldsymbol{\rho}_{I;k=1,K_0}$   is the $K_0$-dimensional vector of air densities in the border region after the final timestep, i.e. for the time of the next measurement;

$\boldsymbol{\rho}_{i;k=1,K_0}$   is the $K_0$-dimensional vector of air densities in the border region at the current timestep as resulting from interpolation in time;

$\boldsymbol{\rho}_{i;j=K_0+1,L_0}$   is the $K_0$-dimensional vector of air densities in the nominal region at the current timestep as resulting from integration according to the MacCormack scheme as described above.

Since the source term depends on air density, the integration in matrix notation for vmr requires simultaneous treatment of vmr and air density, and we get, using notation accordant with air density:

$$
\left( \begin{array}{c} \boldsymbol{\rho}_{i+1} \\ \boldsymbol{vmr}_{i+1} \end{array} \right) = \left( \begin{array}{c} \rho_{l=1} \\ \vdots \\ \rho_{L_0} \\ vmr_{g;l=1} \\ \vdots \\ vmr_{g;\sum L_g} \end{array} \right) = \mathbf{D}_i \left( \begin{array}{c} \boldsymbol{\rho}_i \\ \boldsymbol{vmr}_{g;i} \end{array} \right) = \left( \begin{array}{cccc} \mathbf{D}_{\rho;i} & \mathbf{0} & \mathbf{0} & \mathbf{0} \\ \mathbf{0} & \mathbf{I}_K & \mathbf{0} & \mathbf{0} \\ \mathbf{0} & & \mathbf{W}_i & \mathbf{0} \\ & & \mathbf{D}_{\text{g,nom}} & \end{array} \right) \left( \begin{array}{c} \boldsymbol{\rho}_{i,l=1,L_0} \\ \boldsymbol{vmr}_{g;I,k=1,K_g} \\ \boldsymbol{vmr}_{g;i,k=1,K_g} \\ \boldsymbol{vmr}_{g;i,j=K_g+1,L_g} \end{array} \right) \tag{28}
$$

where $\mathbf{D}_i$ is the total Jacobian with respect to air densities and all involved gas mixing ratios. Note that

1. The Jacobian $\mathbf{D}_i$ is time-dependent because it includes submatrices controlling the interpolation between the initial time and the end time. In the case of vmr, a further time dependence is introduced by the time-dependent source function.

2. the first 'row' of the Jacobian matrix includes identity $\mathbf{I}_K$ because the prediction is not supposed to change the measured $\boldsymbol{\rho}_I$ and $\boldsymbol{vmr}_I$ at the end of the macro time increment. This value is used to construct the boundary condition. Row is here written in quotes because the elements of this 'row' are matrices in themselves. Introduction of unity Jacobian elements is necessary because Eqs. (27–28) are autonomized, originally non-autonomous systems of differential equations.

3. $\mathbf{W}$ is used to interpolate the boundary state between the initial time of the micro time interval, $t + (i-1)\Delta t_p$, and the time at the end of the time interval $t + \Delta t$ to give the atmospheric state at the border region at time $t + i\Delta t_p$.

4. The Jacobian submatrices $\mathbf{D}_{\rho,\text{nom}}$ and $\mathbf{D}_{g,\text{nom}}$ are used to predict the atmospheric state in the nominal range after one further micro time increment from the atmospheric state at the current time and the boundary condition. Its elements are





described in Eqs. (15–21) and (22–26). The part of $\mathbf{D}_{g,\text{nom}}$ which refers to the border mixing ratios ($\boldsymbol{vmr}_{g;I,k=1,K_g}$) is zero. In the case of multiple species, $\mathbf{D}_{\text{g,nom}}$ has a block diagonal structure.

5. No simple mapping mechanism between the field of atmospheric state variables sampled at latitudes and altitudes and the vectors $\boldsymbol{\rho}$ and $\boldsymbol{vmr}_g$ is provided because the fields are irregular in a sense that the number of relevant altitudes is latitude-dependent. Pointer variables have to be used instead.

For the macro time increment $\Delta t$ we get

$$
\begin{pmatrix} \boldsymbol{\rho}_I \\ \boldsymbol{vmr}_I \end{pmatrix} = \left( \prod_{i=I}^{1} \mathbf{D}_i \right) \begin{pmatrix} \boldsymbol{\rho}_0 \\ \boldsymbol{vmr}_0 \end{pmatrix}
\tag{29}
$$

## 3.4 Prediction errors

Let $\mathbf{S}_0$ be the $L \times L$ covariance matrix describing the uncertainties of all involved measurements $\boldsymbol{\rho}_0$ and $\boldsymbol{vmr}_0$, with diagonal elements $s_{0;l,l} = \sigma_{0;l,l}^2$ and $L = \sum_{0}^{gases} L_g$. We assume that these measurement errors in the state variables used for the prediction are the only relevant error sources. With $\mathbf{S}_0$ and $\prod_{i=I}^{1} \mathbf{D}_i$ available, generalized Gaussian error propagation for $\boldsymbol{\rho}_I$ and $\boldsymbol{vmr}_I$ can be easily formulated as:

$$
\mathbf{S}_I = \left( \prod_{i=I}^{1} \mathbf{D}_i \right) \mathbf{S}_0 \left( \prod_{i=I}^{1} \mathbf{D}_i \right)^T .
\tag{30}
$$

Even if $\mathbf{S}_0$ is diagonal, i.e. the initial errors are assumed to be uncorrelated, error propagation through the forward model will generate non-zero error covariances in $\mathbf{S}_I$ representing the atmospheric state at time $t + \Delta t$. $\mathbf{S}_I$ will be needed in the inversion of circulation and mixing coefficients described in Section 4.

## 3.5 A note on finite resolution measurements

The measurements used are not a perfect image of the true atmospheric state but contain some prior information. In the case of the IMK data, a priori profiles are usually set zero, and the constraint is built with a Tikhonov-type first order finite differences smoothing constraint (c.f. von Clarmann et al. (2009). That means that, besides the mapping of measurement and parameter errors, the only distortion of the truth via the retrieval is reduced altitude resolution; no other effect of the prior information is to be considered. Usually, any comparison between modelled and measured fields requires application of the averaging kernels of the retrieval to the model data in order to account for the smoothing by the constraint of the retrieval (assuming that the model grid is much finer than the resolution of the retrieval).

In our case, the situation is different: The model is initialized with measurements of reduced altitude resolution, and the fields predicted by the model are then compared to measurements of the same altitude resolution. It is fair to assume that the model does not dramatically change the altitude resolution of the profiles, and thus comparable quantities are compared when the residuals between predicted and measured atmospheric state are evaluated.





### 3.6 A note on numerical mixing

Let the initial mixing ratio field be homogeneous except one point with delta-type excess mixing ratio. Assume further a homogeneous velocity field and zero mixing coefficients. If the velocity is such that the position of the excess mixing ratio is displaced during $\Delta t$ by a distance which is not equal with an integer multiple of the gridwidth, then the resulting distribution

will no longer be a delta-type distribution but will be smoothed. The widening of the delta peak we refer to as numerical mixing. The MacCormack transport scheme is designed to fight this diffusivity but some higher order effects may still survive. One might think that, during the inversion, the widening is misinterpreted as mixing, leading to too large mixing coefficients.

Again, in our case, the situation is different: The widening does not accumulate over the $\Delta t_p$ timesteps, because we first calculate the operator $\prod_{i=I}^{1} \mathbf{D}_i$, which is applied only once to the initial field, which avoids accumulation of numerical mixing

over timesteps. Still one widening process as described above can occur, when the forward model leads to a position of the new peak which cannot be represented in the grid chosen. However, since the gas distributions $vmr_{g;I}$ at the end of timestep $\Delta t$ are sampled on the same grid, the maximum in the true atmosphere would be widened in the same way, and there would be no residual the inversion would try to get rid of by increasing the mixing coefficients. And the next time step $\Delta t$ is initialized again with measured data, which also excludes accumulation of numerical mixing effects.

These considerations aside, there are other numerical artefacts: These are related to the numerical evaluation of partial derivatives of the state variables in our transport scheme chosen. Particularly in the case of delta functions in the state variable field, these cause side-wiggles behind and smearing in front of the transported structure. To keep these artefacts small, it is necessary to set the spatial grid fine enough that every structure is represented by multiple gridpoints.

## 4 The inverse problem

For convenience, we combine the variables of the initial atmospheric state and the predicted state at the end of the macro time interval, respectively, into the vectors

$$\tilde{\boldsymbol{x}}_0 = \begin{pmatrix} \boldsymbol{\rho}_0 \\ \boldsymbol{vmr}_0 \end{pmatrix} = (\tilde{x}_{0;1} \dots \tilde{x}_{0;L})^T, \tag{31}$$

and

$$\tilde{\boldsymbol{x}}_I = \begin{pmatrix} \boldsymbol{\rho}_I \\ \boldsymbol{vmr}_I \end{pmatrix} = (\tilde{x}_{I;1} \dots \tilde{x}_{I;L})^T, \tag{32}$$

The related subsets of $\tilde{\boldsymbol{x}}_0$ and $\tilde{\boldsymbol{x}}_I$ which contain only state variables in the nominal altitude range but not those in the border region are $\boldsymbol{x}_0 = (x_{0;1} \dots x_{0;J})^T$ and $\boldsymbol{x}_I = (x_{I;1} \dots x_{I;J})^T$, respectively. The reason why the distinction between $\tilde{\boldsymbol{x}}$ and $\boldsymbol{x}$ is made is that, contrary to the prediction step, for the inversion vector elements related to the interpolation of values in the border region are no longer needed. Further, we combine the fields of meridional and vertical wind components and mixing





coefficients into the vector

$$q = \begin{pmatrix} v \\ w \\ K_\phi \\ K_z \end{pmatrix}, \tag{33}$$

and assume constant velocities and mixing coefficients during the macro timestep. To infer circulation patterns and mixing coefficients from the measurements of air densities and mixing ratios, the Jacobian matrix $\mathbf{F}$,

$$\mathbf{F} = (\boldsymbol{f}_1, \dots, \boldsymbol{f}_N) = (f_{j,n}) = \left( \frac{\partial x_{I,j}}{\partial q_n} \right) = \left( \frac{\partial \boldsymbol{x}_I}{\partial q_1}, \dots, \frac{\partial \boldsymbol{x}_I}{\partial q_N} \right), \tag{34}$$

is needed, where $N = 4J$ where $J = \sum_0^{gases} J_g$, because there are four unknown quantities, $v_j$, $w_j$, $K_\phi$, and $K_z$ at each gridpoint of the nominal region where these variables shall be inferred. The elements of $\mathbf{F}$ are calculated from Eq. (28) by application of the product rule:

$$\tilde{\boldsymbol{f}}_n = \sum_{i=1}^{I} \left[ \left( \prod_{k=I}^{i+1} \mathbf{D}_k \right) \left( \frac{\partial \mathbf{D}_i}{\partial q_n} \right) \left( \prod_{k=i-1}^{1} \mathbf{D}_k \right) \tilde{\boldsymbol{x}}_0 \right], \tag{35}$$

where the tilde symbol in $\tilde{\boldsymbol{f}}_n$ indicates that the vectors resulting from Eq. (35) still include the border elements which still have to be discarded to obtain $\boldsymbol{f}_n$. The quantity $\tilde{\boldsymbol{f}}_n$ is more efficiently computed using the following recursive scheme, where $\tilde{\boldsymbol{f}}_{l,i}$ is the respective column of the Jacobian after micro timestep $i$:

$$\tilde{\boldsymbol{f}}_{n,i} = \mathbf{D}_i \boldsymbol{f}_{n,i-1} + \frac{\partial \mathbf{D}_i}{\partial q_n} \left( \prod_{k=i-1}^{1} \mathbf{D}_k \right) \tilde{\boldsymbol{x}}_0 \tag{36}$$

With the argument of $\mathbf{D}$ specifying the column of the $\mathbf{D}$-matrix such that $D_{c,i}(\phi,z)$ relates $\rho_{i+1}(\phi,z)$ to $\rho_i(\phi,z)$, $D_{\rho,i}(\phi \pm \Delta\phi, z)$ relates $\rho_{i+1}(\phi,z)$ to $\rho_i(\phi \pm \Delta\phi, z)$, and $D_{\rho,i}(\phi, z \pm \Delta z)$ relates $\rho_{i+1}(\phi,z)$ to $\rho_i(\phi, z \pm \Delta z)$, and for vmr accordingly, the entries of $\mathbf{D}_i$ relevant to $v$ are:

$$\frac{\partial \frac{\partial \rho_{i+1}(\phi,z)}{\partial \rho_i(\phi,z)}}{\partial v(\phi,z)} = -\frac{\Delta t_p}{2\Delta\phi} \cdot \left( \frac{\Delta t_p}{\Delta\phi} \cdot \frac{2v(\phi,z) + v(\phi - \Delta\phi, z)}{r^2} + 2\frac{\Delta t_p}{\Delta z} \cdot \frac{w(\phi,z)}{r} \right) \tag{37}$$

$$\frac{\partial \frac{\partial \rho_{i+1}(\phi+\Delta\phi,z)}{\partial \rho_i(\phi+\Delta\phi,z)}}{\partial v(\phi,z)} = -\frac{1}{2} \cdot \left( \frac{\Delta t_p}{\Delta\phi} \right)^2 \cdot \frac{v(\phi+\Delta\phi,z)}{r^2} \tag{38}$$

$$\frac{\partial \frac{\partial \rho_{i+1}(\phi+\Delta\phi,z)}{\partial \rho_i(\phi,z)}}{\partial v(\phi,z)} = \frac{1}{2r} \cdot \frac{\Delta t_p}{\Delta\phi} \cdot \frac{\cos(\phi)}{\cos(\phi+\Delta\phi)} \cdot \left( 1 + 2\frac{\Delta t_p}{\Delta\phi} \cdot \frac{v(\phi,z)}{r} + \frac{\Delta t_p}{\Delta z} w(\phi,z) \right) \tag{39}$$

$$\frac{\partial \frac{\partial \rho_{i+1}(\phi,z)}{\partial \rho_i(\phi+\Delta\phi,z)}}{\partial v(\phi,z)} = \frac{1}{2} \cdot \left( \frac{\Delta t_p}{\Delta\phi} \right)^2 \cdot \frac{v(\phi+\Delta\phi,z)}{r^2} \cdot \frac{\cos(\phi+\Delta\phi)}{\cos(\phi)} \tag{40}$$





$$\frac{\partial \frac{\partial \rho_{i+1}(\phi-\Delta\phi,z)}{\partial \rho_i(\phi,z)}}{\partial v(\phi,z)} = \frac{1}{2r} \cdot \frac{\Delta t_p}{\Delta\phi} \cdot \frac{\cos(\phi)}{\cos(\phi-\Delta\phi)} \cdot \left( -1 + \frac{\Delta t_p}{\Delta\phi} \cdot \frac{v(\phi-\Delta\phi,z)}{r} + \frac{\Delta t_p}{\Delta z} w(\phi-\Delta\phi,z) \right) \tag{41}$$

$$\frac{\partial \frac{\partial \rho_{i+1}(\phi,z+\Delta z)}{\partial \rho_i(\phi,z)}}{\partial v(\phi,z)} = \frac{1}{2} \cdot \frac{\Delta t_p}{\Delta z} \cdot \frac{\Delta t_p}{\Delta\phi} \cdot \frac{r^2}{(r+\Delta z)^2} \frac{w(\phi,z)}{r} \tag{42}$$

$$\frac{\partial \frac{\partial \rho_{i+1}(\phi,z)}{\partial \rho_i(\phi,z+\Delta z)}}{\partial v(\phi,z)} = \frac{1}{2} \cdot \frac{\Delta t_p}{\Delta z} \cdot \frac{\Delta t_p}{\Delta\phi} \cdot \frac{(r+\Delta z)^2}{r^3} w(\phi,z+\Delta z) \tag{43}$$

$$\frac{\partial \frac{\partial \rho_{i+1}(\phi+\Delta\phi,z)}{\partial \rho_i(\phi,z+\Delta z)}}{\partial v(\phi,z)} = -\frac{1}{2} \cdot \frac{\Delta t_p}{\Delta\phi} \cdot \frac{\Delta t_p}{\Delta z} \cdot \frac{(r+\Delta z)^2}{r^3} \cdot \frac{\cos(\phi)}{\cos(\phi+\Delta\phi)} w(\phi,z+\Delta z) \tag{44}$$

$$\frac{\partial \frac{\partial \rho_{i+1}(\phi-\Delta\phi,z+\Delta z)}{\partial \rho_i(\phi,z)}}{\partial v(\phi,z)} = -\frac{1}{2} \cdot \frac{\Delta t_p}{\Delta\phi} \cdot \frac{\Delta t_p}{\Delta z} \cdot \frac{r}{(r+\Delta z)^2} \cdot \frac{\cos(\phi)}{\cos(\phi-\Delta\phi)} \cdot w(\phi-\Delta\phi,z) \tag{45}$$

$$\frac{\partial \frac{\partial vmr_{i+1}(\phi,z)}{\partial vmr_i(\phi,z)}}{\partial v(\phi,z)} = -\left(\frac{\Delta t_p}{\Delta\phi}\right)^2 \cdot \frac{1}{r^2} \cdot \left( v(\phi,z) + \frac{v(\phi-\Delta\phi,z)}{2} \right) \tag{46}$$

$$\frac{\partial \frac{\partial vmr_{i+1}(\phi+\Delta\phi,z)}{\partial vmr_i(\phi+\Delta\phi,z)}}{\partial v(\phi,z)} = -\left(\frac{\Delta t_p}{\Delta\phi}\right)^2 \cdot \frac{v(\phi+\Delta\phi,z)}{2r^2} \tag{47}$$

$$\frac{\partial \frac{\partial vmr_{i+1}(\phi,z)}{\partial vmr_i(\phi-\Delta\phi,z)}}{\partial v(\phi,z)} = \frac{1}{2r} \cdot \frac{\Delta t_p}{\Delta\phi} \cdot \left( 1 + \frac{\Delta t_p}{\Delta\phi} \cdot \frac{v(\phi-\Delta\phi,z)}{r} \right) \tag{48}$$

$$\frac{\partial \frac{\partial vmr_{i+1}(\phi+\Delta\phi,z)}{\partial vmr_i(\phi,z)}}{\partial v(\phi,z)} = \frac{1}{2r}\left(\frac{\Delta t_p}{\Delta\phi}\right)^2 \frac{v(\phi+\Delta\phi,z)}{r} \tag{49}$$

$$\frac{\partial \frac{\partial vmr_{i+1}(\phi,z)}{\partial vmr_i(\phi+\Delta\phi,z)}}{\partial v(\phi,z)} = -\frac{\Delta t_p}{\Delta\phi} \cdot \frac{1}{r}\left( \frac{1}{2} - \frac{\Delta t_p}{\Delta\phi} \cdot \frac{v(\phi,z)}{r} \right) \tag{50}$$

Entries not mentioned here are zero. Entries relevant to $w$ are:

$$\frac{\partial \frac{\partial \rho_{i+1}(\phi,z)}{\partial \rho_i(\phi,z)}}{\partial w(\phi,z)} = -\frac{\Delta t_p}{\Delta\phi} \cdot \frac{\Delta t_p}{\Delta z} \cdot \frac{v(\phi,z)}{r} - \left(\frac{\Delta t_p}{\Delta z}\right)^2 w(\phi,z) - \frac{1}{2}\left(\frac{\Delta t_p}{\Delta z}\right)^2 w(\phi,z-\Delta z) \tag{51}$$

$$\frac{\partial \frac{\partial \rho_{i+1}(\phi,z+\Delta z)}{\partial \rho_i(\phi,z+\Delta z)}}{\partial w(\phi,z)} = -\frac{1}{2}\left(\frac{\Delta t_p}{\Delta z}\right)^2 w(\phi,z+\Delta z) \tag{52}$$

$$\frac{\partial \frac{\partial \rho_{i+1}(\phi,z)}{\partial \rho_i(\phi+\Delta\phi,z)}}{\partial w(\phi,z)} = \frac{1}{2} \cdot \frac{\Delta t_p}{\Delta\phi} \cdot \frac{\Delta t_p}{\Delta z} \cdot \frac{v(\phi+\Delta\phi,z)}{r} \cdot \frac{\cos(\phi+\Delta\phi)}{\cos(\phi)} \tag{53}$$





$$\frac{\partial \frac{\partial \rho_{i+1}(\phi+\Delta\phi,z)}{\partial \rho_i(\phi+\Delta\phi,z-\Delta z)}}{\partial w(\phi,z)} = \frac{1}{2} \cdot \frac{\Delta t_p}{\Delta\phi} \cdot \frac{\Delta t_p}{\Delta z} \cdot \frac{v(\phi,z)}{r} \cdot \frac{\cos(\phi)}{\cos(\phi+\Delta\phi)} \tag{54}$$

$$\frac{\partial \frac{\partial \rho_{i+1}(\phi,z+\Delta z)}{\partial \rho_i(\phi,z)}}{\partial w(\phi,z)} = \frac{1}{2} \cdot \frac{\Delta t_p}{\Delta z} \cdot \frac{r^2}{(r+\Delta z)^2} \cdot \left(1 + \frac{\Delta t_p}{\Delta\phi} \cdot \frac{v(\phi,z)}{r} + 2\frac{\Delta t_p}{\Delta z} \cdot w(\phi,z)\right) \tag{55}$$

$$\frac{\partial \frac{\partial \rho_{i+1}(\phi,z)}{\partial \rho_i(\phi,z+\Delta z)}}{\partial w(\phi,z)} = \frac{1}{2} \cdot \left(\frac{\Delta t_p}{\Delta z}\right)^2 \cdot \frac{(r+\Delta z)^2}{r^2} \cdot w(\phi,z+\Delta z) \tag{56}$$

$$\frac{\partial \frac{\partial \rho_{i+1}(\phi,z-\Delta z)}{\partial \rho_i(\phi,z)}}{\partial w(\phi,z)} = \frac{1}{2} \cdot \frac{\Delta t_p}{\Delta z} \cdot \frac{r^2}{(r-\Delta z)^2} \cdot \left(-1 + \frac{\Delta t_p}{\Delta\phi} \cdot \frac{v(\phi,z-\Delta z)}{r-\Delta z} + \frac{\Delta t_p}{\Delta z} \cdot w(\phi,z-\Delta z)\right) \tag{57}$$

$$\frac{\partial \frac{\partial \rho_{i+1}(\phi+\Delta\phi,z-\Delta z)}{\partial \rho_i(\phi,z)}}{\partial w(\phi,z)} = -\frac{1}{2} \cdot \frac{\Delta t_p}{\Delta\phi} \cdot \frac{\Delta t_p}{\Delta z} \cdot \frac{r^2}{(r-\Delta z)^2} \cdot \frac{\cos(\phi)}{\cos(\phi+\Delta\phi)} \frac{v(\phi,z-\Delta z)}{r-\Delta z} \tag{58}$$

$$\frac{\partial \frac{\partial \rho_{i+1}(\phi,z+\Delta z)}{\partial \rho_i(\phi+\Delta\phi,z)}}{\partial w(\phi)} = -\frac{1}{2} \cdot \frac{\Delta t_p}{\Delta\phi} \cdot \frac{\Delta t_p}{\Delta z} \cdot \frac{r^2}{(r+\Delta z)^2} \cdot \frac{\cos(\phi+\Delta\phi)}{\cos(\phi)} \cdot \frac{v(\phi+\Delta\phi,z)}{r} \tag{59}$$

$$\frac{\partial \frac{\partial vmr_{i+1}(\phi,z)}{\partial vmr_i(\phi,z)}}{\partial w(\phi,z)} = -\left(\frac{\Delta t_p}{\Delta z}\right)^2 \cdot \left(w(\phi,z) + \frac{w(\phi,z-\Delta z)}{2}\right) \tag{60}$$

$$\frac{\partial \frac{\partial vmr_{i+1}(\phi,z+\Delta z)}{\partial vmr_i(\phi,z+\Delta z)}}{\partial w(\phi,z)} = -\left(\frac{\Delta t_p}{\Delta z}\right)^2 \cdot \frac{w(\phi,z+\Delta z)}{2} \tag{61}$$

$$\frac{\partial \frac{\partial vmr_{i+1}(\phi,z)}{\partial vmr_i(\phi,z-\Delta z)}}{\partial w(\phi,z)} = \frac{1}{2} \cdot \frac{\Delta t_p}{\Delta z} \cdot \left(1 + w(\phi,z-\Delta z) \cdot \frac{\Delta t_p}{\Delta z}\right) \tag{62}$$

$$\frac{\partial \frac{\partial vmr_{i+1}(\phi,z+\Delta z)}{\partial vmr_i(\phi,z)}}{\partial w(\phi,z)} = \frac{1}{2} \cdot \left(\frac{\Delta t_p}{\Delta z}\right)^2 \cdot w(\phi,z+\Delta z) \tag{63}$$

$$\frac{\partial \frac{\partial vmr_{i+1}(\phi,z)}{\partial vmr_i(\phi,z+\Delta z)}}{\partial w(\phi,z)} = -\frac{1}{2} \cdot \frac{\Delta t_p}{\Delta z} \cdot \left(1 - 2\frac{(\Delta t_p)}{(\Delta z)} \cdot w(\phi,z)\right) \tag{64}$$

Entries relevant to $\boldsymbol{K}_\phi$ are:

$$\frac{\partial \frac{\partial vmr_{i+1}(\phi,z)}{\partial vmr_i(\phi,z)}}{\partial K_\phi(\phi,z)} = -\frac{\Delta t_p}{(\Delta\phi)^2} \cdot \frac{1}{2r^2} \cdot \frac{\cos\left(\phi+\frac{\Delta\phi}{2}\right) + \cos\left(\phi-\frac{\Delta\phi}{2}\right)}{\cos(\phi)} \tag{65}$$

$$\frac{\partial \frac{\partial vmr_{i+1}(\phi\mp\Delta\phi,z)}{\partial vmr_i(\phi\mp\Delta\phi,z)}}{\partial K_\phi(\phi,z)} = -\frac{\Delta t_p}{(\Delta\phi)^2} \cdot \frac{1}{2r^2} \cdot \frac{\cos\left(\phi\mp\frac{\Delta\phi}{2}\right)}{\cos(\phi\mp\Delta\phi)} \tag{66}$$

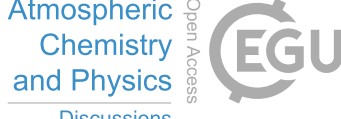



$$\partial\frac{\frac{\partial vmr_{i+1}(\phi,z)}{\partial vmr_i(\phi-\Delta\phi,z)}}{\partial K_\phi(\phi,z)} = \frac{1}{2r^2} \cdot \frac{\Delta t_p}{(\Delta\phi)^2} \cdot \frac{\cos(\phi-\frac{\Delta\phi}{2})}{\cos(\phi)} \tag{67}$$

$$\partial\frac{\frac{\partial vmr_{i+1}(\phi,z)}{\partial vmr_i(\phi+\Delta\phi,z)}}{\partial K_\phi(\phi,z)} = \frac{1}{2r^2} \cdot \frac{\Delta t_p}{(\Delta\phi)^2} \cdot \frac{\cos(\phi+\frac{\Delta\phi}{2})}{\cos(\phi)} \tag{68}$$

$$\partial\frac{\frac{\partial vmr_{i+1}(\phi+\Delta\phi,z)}{\partial vmr_i(\phi,z)}}{\partial K_\phi(\phi,z)} = \frac{1}{2r^2} \cdot \frac{\Delta t_p}{(\Delta\phi)^2} \cdot \frac{\cos(\phi+\frac{\Delta\phi}{2})}{\cos(\phi+\Delta\phi)} \tag{69}$$

$$\partial\frac{\frac{\partial vmr_{i+1}(\phi-\Delta\phi,z)}{\partial vmr_i(\phi,z)}}{\partial K_\phi(\phi,z)} = \frac{1}{2r^2} \cdot \frac{\Delta t_p}{(\Delta\phi)^2} \cdot \frac{\cos(\phi-\frac{\Delta\phi}{2})}{\cos(\phi-\Delta\phi)} \tag{70}$$

And finally, entries relevant to $\boldsymbol{K}_z$ are:

$$\partial\frac{\frac{\partial vmr_{i+1}(\phi,z)}{\partial vmr_i(\phi,z)}}{\partial K_z(\phi,z)} = -\frac{\Delta t_p}{(\Delta z)^2} \cdot \frac{(r+\frac{\Delta z}{2})^2 + (r-\frac{\Delta z}{2})^2}{2r^2} \tag{71}$$

$$\partial\frac{\frac{\partial vmr_{i+1}(\phi,z\mp\Delta z)}{\partial vmr_i(\phi,z\mp\Delta z)}}{\partial K_z(\phi,z)} = -\frac{\Delta t_p}{(\Delta z)^2} \cdot \frac{(r\mp\frac{\Delta z}{2})^2}{2(r\mp\Delta z)^2} \tag{72}$$

$$\partial\frac{\frac{\partial vmr_{i+1}(\phi,z+\Delta z)}{\partial vmr_i(\phi,z)}}{\partial K_z(\phi,z)} = \frac{1}{2} \cdot \frac{\Delta t_p}{(\Delta z)^2} \cdot \frac{(r+\frac{\Delta z}{2})^2}{(r+\Delta z)^2} \tag{73}$$

$$\partial\frac{\frac{\partial vmr_{i+1}(\phi,z)}{\partial vmr_i(\phi,z-\Delta z)}}{\partial K_z(\phi,z)} = \frac{1}{2} \cdot \frac{\Delta t_p}{(\Delta z)^2} \cdot \frac{(r-\frac{\Delta z}{2})^2}{r^2} \tag{74}$$

$$\partial\frac{\frac{\partial vmr_{i+1}(\phi,z)}{\partial vmr_i(\phi,z+\Delta z)}}{\partial K_z(\phi,z)} = \frac{1}{2} \cdot \frac{\Delta t_p}{(\Delta z)^2} \cdot \frac{(r+\frac{\Delta z}{2})^2}{r^2} \tag{75}$$

$$\partial\frac{\frac{\partial vmr_{i+1}(\phi,z-\Delta z)}{\partial vmr_i(\phi,z)}}{\partial K_z(\phi,z)} = \frac{1}{2} \cdot \frac{\Delta t_p}{(\Delta z)^2} \cdot \frac{(r-\frac{\Delta z}{2})^2}{(r-\Delta z)^2} \tag{76}$$

We linearize the prediction with respect to wind and mixing coefficients

$$\boldsymbol{x}_I = \boldsymbol{x}_0 + \mathbf{F}(\boldsymbol{q} - \boldsymbol{q}_0). \tag{77}$$

Assuming linearity and Gaussian statistics, the most likely set $\boldsymbol{q}$ of winds and mixing ratios during the macro time interval minimizes the following cost function:

$$
\begin{aligned}
\chi_1^2 &= (\boldsymbol{x}_{m;I} - \boldsymbol{x}_I)^T \mathbf{S}_r^{-1} (\boldsymbol{x}_{m;I} - \boldsymbol{x}_I) \\
&\approx (\boldsymbol{x}_{m;I} - \boldsymbol{x}_0 - \mathbf{F}(\boldsymbol{q} - \boldsymbol{q}_0))^T \mathbf{S}_r^{-1} (\boldsymbol{x}_{m;I} - \boldsymbol{x}_0 - \mathbf{F}(\boldsymbol{q} - \boldsymbol{q}_0))
\end{aligned}
\tag{78}
$$





where $x_{\mathrm{m};I}$ is the measured state at the end of the macro time step and $\mathbf{S}_{\mathrm{r}}$ is the error covariance matrix of the residual, which is, under the assumption that prediction error and measurement errors are uncorrelated, the sum of the prediction covariance matrix and the measurement covariance matrix, both after the macro time step:

$$\mathbf{S}_{\mathrm{r}} = \mathbf{S}_{\mathrm{m},I} + \mathbf{S}_{\mathrm{p}}, \tag{79}$$

where $\mathbf{S}_{\mathrm{p}}$ is an $J \times J$-matrix containing those elements of $\mathbf{S}_I$ which are relevant to $x_I$. $\mathbf{S}_{\mathrm{m},I}$ is the measurement error covariance matrix of the atmospheric state after the macro time step. The minimization of the cost function gives the following estimate $\hat{q}$ of winds and mixing coefficients:

$$\hat{q} = q_0 + \left(\mathbf{F}^T \mathbf{S}_{\mathrm{r}}^{-1} \mathbf{F}\right)^{-1} \mathbf{F}^T \mathbf{S}_{\mathrm{r}}^{-1} (x_{\mathrm{m};I} - x_I) \tag{80}$$

The matrix $\mathbf{F}^T \mathbf{S}_{\mathrm{r}}^{-1} \mathbf{F}$ can be singular either because the related system of equations is under-determined or ill-posed due to nearly linearly dependent equations. Singularity is fought by adding the following constraint term to the cost function of Eq. (78):

$$\chi^2_{\mathrm{con}} = (q - q_a)^T \mathbf{R} (q - q_a) \tag{81}$$
$$\chi^2 = \chi^2_1 + \chi^2_{\mathrm{con}}, \tag{82}$$

where $q_a$ is some prior assumption on velocities and mixing coefficients. $\mathbf{R}$ is a $J \times J$ regularization matrix of which the choice is discussed below. From this, the constrained estimate of velocities and mixing coefficients can be inferred:

$$\hat{q} = q_a + \left(\mathbf{F}^T \mathbf{S}_{\mathrm{r}}^{-1} \mathbf{F} + \mathbf{R}\right)^{-1} \mathbf{F}^T \mathbf{S}_{\mathrm{r}}^{-1} (x_{\mathrm{m};I} - x_I) \tag{83}$$

An equivalent formulation, which is more efficient if the dimension of $q$ is larger than that of $x$ (underdetermined problem), but which requires a non-singular regularization matrix, and which does not give easy access to diagnostics (see below), is (Rodgers, 2000):

$$\hat{q} = q_a + \mathbf{R}^{-1} \mathbf{F}^T \left(\mathbf{F} \mathbf{R}^{-1} \mathbf{F}^T + \mathbf{S}_{\mathrm{r},I}\right)^{-1} (x_{\mathrm{m};I} - x_I). \tag{84}$$

The covariance matrix characterizing the uncertainty of estimated winds and mixing coefficients is

$$\mathbf{S}_q = \left(\mathbf{F}^T \mathbf{S}_{\mathrm{r}}^{-1} \mathbf{F} + \mathbf{R}\right)^{-1} \mathbf{F}^T \mathbf{S}_{\mathrm{r}}^{-1} \mathbf{F} \left(\mathbf{F}^T \mathbf{S}_{\mathrm{r}}^{-1} \mathbf{F} + \mathbf{R}\right)^{-1}, \tag{85}$$

and the estimated winds and mixing coefficients are related to the true ones as

$$\mathbf{A} = \frac{\partial \hat{q}}{\partial q} = \left(\mathbf{F}^T \mathbf{S}_{\mathrm{r}}^{-1} \mathbf{F} + \mathbf{R}\right)^{-1} \mathbf{F}^T \mathbf{S}_{\mathrm{r}}^{-1} \mathbf{F}, \tag{86}$$

which is unity in the case of unconstrained estimation of $q$. In the case of Newtonian iteration, Eqs. (85-86) are evaluated using the Jacobian $\mathbf{F}$ valid at the solution.





Due to the concentration-dependence of the source function and the $q$-depecdence of $\mathbf{F}$, Eq. (29) is valid only in linear approximation. This is helped by putting the inversion in the context of a Newtonian iteration (see, e.g., Rodgers (2000, p. 85)). Eq. (80) becomes

$$\hat{\boldsymbol{q}}_{it+1} = \boldsymbol{q}_{it} + \left(\mathbf{F}_{it}^T \mathbf{S}_{\mathrm{r}}^{-1} \mathbf{F}_{it}\right)^{-1} \mathbf{F}_{it}^T \mathbf{S}_{\mathrm{r}}^{-1} (\boldsymbol{x}_{\mathrm{m};I} - \boldsymbol{x}_{I,it}), \tag{87}$$

where $it$ is the iteration index. Equation (83) becomes

$$\hat{\boldsymbol{q}}_{it+1} \;\; = \;\; \boldsymbol{q}_{it} + \left(\mathbf{F}_{it}^T \mathbf{S}_{\mathrm{r}}^{-1} \mathbf{F}_{it} + \mathbf{R}\right)^{-1} \left(\mathbf{F}_{it}^T \mathbf{S}_{\mathrm{r}}^{-1} (\boldsymbol{x}_{\mathrm{m};I} - \boldsymbol{x}_{I,it}) - \mathbf{R}(\boldsymbol{q}_{it} - \boldsymbol{q}_a)\right) \tag{88}$$

or alternatively

$$\hat{\boldsymbol{q}}_{it+1} \;\; = \;\; \boldsymbol{q}_a + \left(\mathbf{F}_{it}^T \mathbf{S}_{\mathrm{r}}^{-1} \mathbf{F}_{it} + \mathbf{R}\right)^{-1} \mathbf{F}_{it}^T \mathbf{S}_{\mathrm{r}}^{-1} \left(\boldsymbol{x}_{\mathrm{m};I} - \boldsymbol{x}_{I,it} + \mathbf{F}_{it}(\boldsymbol{q}_{it} - \boldsymbol{q}_a)\right), \tag{89}$$

and Eq. (84) becomes

$$\hat{\boldsymbol{q}}_{it+1} \;\; = \;\; \boldsymbol{q}_a + \mathbf{R}^{-1} \mathbf{F}_{it}^T \left(\mathbf{F}_{it} \mathbf{R}^{-1} \mathbf{F}_{it}^T + \mathbf{S}_{\mathrm{r},I}\right)^{-1} \left(\boldsymbol{x}_{\mathrm{m};I} - \boldsymbol{x}_{I,it} + \mathbf{F}_{it}(\boldsymbol{q}_{it} - \boldsymbol{q}_a)\right). \tag{90}$$

With $\boldsymbol{q}_a = \mathbf{0}$ and diagonal $\mathbf{R} = \gamma\mathbf{I}$ we get the smallest possible velocities and mixing coefficients still consistent with the measurement, where tuning parameter $\gamma$ will be set depending on how large fit residuals the user still considers to be 'consistent'. With $\mathbf{R}$ being diagonally blockwise composed of squared and scaled first order finite differences operators and $\boldsymbol{q}_a = \mathbf{0}$, smooth fields of wind vectors and mixing coefficients can be enforced. Setting $\boldsymbol{q}_a$ the result of the previous macro time step and $\mathbf{R}$ its reciprocal uncertainty plus some margin for allowance of variability of velocity and mixing coefficients in time corresponds to sequential data assimilation. And finally, if prior knowledge is formed by independent measurements and their reciprocal uncertainties as constraint matrix, or within the debatable framework of Bayesian statistics, estimates $\hat{\boldsymbol{q}}$ would even be the most probable estimate of velocities and mixing ratios.

## 5 Proof of concept

### 5.1 Prediction of the atmospheric state

In a first step we test the predictive power of the formalism defined by Eqs. (3–29). Since the formalism itself is deductive and starts from a well established theoretical concept, the purpose of the test is solely to verify that the implementation of the formalism is correct and that involved numerical approximations are adequate. As a consequence of the Bonini paradox (c.f. Bonini (1963) and Starbuck (1975)), a model is the harder to understand the more complex it is. While the predictive power of a model usually increases with complexity, this does not necessarily hold for its explanatory power. Thus we have decided to test our model on the basis of very simple test cases, where major failure of the model is immediately obvious. Four test cases have been chosen, each dedicated to one kinematic variable ($v$, $w$, $K_\phi$ and $K_z$), while the other three were set zero.

In the first case, $v$ was set close to the Courant limit (Courant et al., 1928) (about $0.17\cos(\phi)$ ms$^{-1}$) everywhere. As one would expect from the continuity equation applied to a spherical atmosphere, no changes in air density except boundary effects





at the poles were observed, and structures near the equator were transported by about $4°$ within a month, as expected from the equation of motion. A Gaussian-shaped perturbation of a halfwidth of one latitudinal gridwidth ($4°$) causes an upwind wiggle of less than 0.7% of the amplitude of the perturbation at a meridional velocity of one gridpoint per month. There is no discernable change in the width of the transported structure. Similarly, for the second case a constant field of $w$ of $1.1 \times 10^{-3}$

$ms^{-1}$ lifts a structure upwards by about 3 km per month. Mixing coefficients were verified to smear out structures in the respective direction while leaving air density and structures in the orthogonal direction unchanged.

## 5.2   Inversion of simulated measurements

Case studies based on real measurement data are inadequate as the sole proof of concept because the truth is unknown and the result thus is unverifiable. Instead we first test our scheme on the basis of simulated atmospheric states and consider the scheme

as verified if the velocities and mixing coefficients used to simulate the atmospheric states are sufficiently well reproduced. In the noise-free well conditioned case one might even expect, within the numerical precision of the system, the exact reproduction of the reference data; due to the – weak but non-zero – dispersivity of the numerical transport scheme the wiggles discussed in the previous subsection cause, at some gridpoints, $\mathbf{D}$-matrix entries of the wrong sign. In order to fight resulting convergence problems of the inversion, at least some small regularization is adequate, even if the system of equations to be solved is well or

over-determined. Since the system in reality is, in tendency, ill-conditioned and the constraint applied to the inversion prevents reproduction of the reference data, we use a variety of idealized tracers instead. After this initial test of functionality, more and more realistic test cases are constructed in order to study the competing influence of constraint and measurement data on the solution.

The trace gas distributions used for this test were chosen such that the rows of Equation (80) are independent. Four artificial

gases were chosen whose mixing ratios had a linear, quadratic and two variants of exponential latitude and altitude dependences, respectively. Further, a hydrostatic air density distribution was assumed. After successful separate inversion of $v_{phi}$, $v_z$, $K_{phi}$ and $K_z$, these kinematic variables were inverted in combination, whereby no further problems were encountered.

In a following step, the linear latitude-dependence was replaced by a stepwise linear function, i.e., a dependence on the absolute amount of latitude. Here it showed up that, besides ill-posedness due to linear dependence of equations, the unphysical

upwind wiggles in the vicinity of the mixing ratio peak as discussed in the previous subsection can trigger errors which are boosted during the iteration. This problem, which is associated with sharp structures and large velocities (of the order of one gridwidth per macrotimestep) can be solved by the use of a smoothing regularization matrix $\mathbf{R}$ as discussed in the last paragraph of Section 4, however at the cost of degraded spatial resolution of the result.

## 5.3   Case Study with MIPAS measurements

The risk of case studies based on simulated data typically is that not all difficulties encountered with real data are foreseen during theoretical studies. In order to demonstrate applicability to real data, global monthly latitude/altitude distributions of CFC-12, $CH_4$, $N_2O$ and $SF_6$ (Kellmann et al., 2012; Plieninger et al., 2015; Haenel et al., 2015) measured with the Michelson Interferometer for Passive Atmospheric Sounding (MIPAS) (Fischer et al., 2008) were used. The purpose of these tests is





demonstration of the feasibility of the method presented. An investigation of the atmospheric circulation on the basis of this method applied to MIPAS data is left for a companion paper. For this proof of concept, sinks of these long-lived tracers have been ignored but these will certainly be considered in scientific applications.

For this case study, zonal monthly mean distributions of air densities and mixing ratios of these four species from September and October 2010 were used. Figure 1 shows the measured distributions of these quantities in September (left column) and October (middle column), and the residuals between the measured and predicted contributions for October (right column). The residuals are reasonably small and show, except for methane in the polar upper stratosphere, no patterns which would hint at peculiarities with the inferred kinematic quantities.

The resulting circulation vectors which best explain the change of the mixing ratio distributions from September to October 2010 are shown in the upper left panel of Figure 2. Winter polar subsidence, summer polar upwelling, the mesospheric over-turning circulation, the upper and lower branches of the Brewer-Dobson circulation and the tropical pipe are clearly visible. Details of the tropical pipe are visible in the right upper panel. As expected, the Brewer-Dobson circulation is much more pronounced in the northern (early) winter hemisphere. Velocities are roughly consistent with mean ages of stratospheric air as determined by Stiller et al. (2008) and Haenel et al. (2015). While the inferred field of circulation vectors shows many detail features demanding scientific investigation in their own right, the reproduction of the expected features justifies confidence in the method proposed. Resulting mixing coefficients $K_\phi$ and $K_z$ are shown in the left and right lower panels, respectively. Negative mixing ratios indicate counter-gradient mixing, which seems to be most pronounced in the tropical upper stratosphere.

Jacobian elements with respect to $v$ values and $K$ values seem to form a null space. Thus the $K$-values were constrained to zero using diagonal components only in the $\mathbf{R}$ matrix diagonal blocks associated with the $K$ values. The strength of this constraint was adjusted such that the $K$ values were as small as possible as long as this did not boost the residual. Resulting $K_\phi$ and $K_z$ distributions are shown in Figure 3.

The errors in the estimated transport velocities and mixing coefficients have been estimated according to Equation (85) and are shown in Figure 3, middle column. The errors in the transport velocities are in the one percent range, indicating that the information contained in the measurements is adequate for the purpose of retrieving circulation parameters. It seems even possible to improve the time resolution of the circulation analysis and aim at weakly instead of monthly temporal sampling.

Larger errors above 65 km altitude and at the bins closest to the pole are border effects, resulting from the fact that no symmetric derivatives can be calculated there. The uncertainties in $K_\phi$ show the same patterns as the $K_\phi$ values themselves.

## 6 Discussion

The analysis of the age of stratospheric air can be understood as an integrated view at the equations of motion of stratospheric air, because the total travel time of the air parcel through the stratosphere is represented. The refinement of this method which analyzes the mean age just considers a weighted mean of the above, but it is still an integral method. Contrary to these integral methods, our direct inversion scheme supports a – in approximation, due to discrete sampling in the time domain – differential view at the same problem. The related advantages are: (a) independence of assumptions on the age spectrum,



**Figure 1.** : Measured distributions in September (left column), October (middle column) and residual distributions between October measurements and predictions for October (right column) for air density and mixing ratios of CFC-12, $CH_4$, $N_2O$, and $SF_6$ (top to bottom). Grey gridboxes indicate non-availability of valid data.





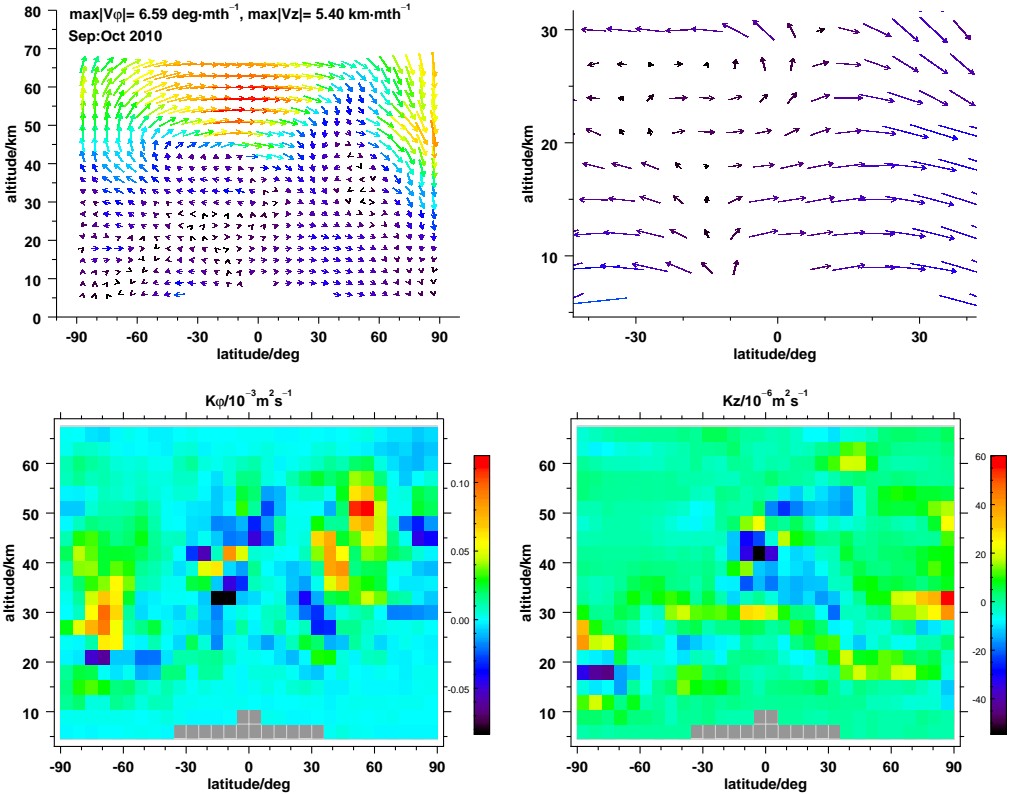

**Figure 2.** : Resulting circulation vectors $(v_\phi(z,\phi); v_z(z,\phi))$ (upper left panel), where colours on the red side of the rainbow colour scale represent higher velocities; a detail of this (upper right panel); mixing coefficients $K_{phi}$ (lower left panel) and $K_z$ (lower right panel).

because during each time step mixing is explicitly considered; (b) insensitivity to $SF_6$ depletion in the mesosphere (c.f., e.g., Reddmann et al., 2001; Stiller et al., 2012), because the scheme uses the actual entry values of subsiding air as a reference; (c) applicability to non-ideal tracers in the stratosphere; since the atmospheric state is updated for each time step by measured value, depletion does not accumulate, even if no sink functions are considered; and (d) the logical circle that the lifetimes of

5 non-ideal tracers depend on their trajectories (and thus atmospheric circulation), while the determination of the circulation requires knowledge of the lifetimes, can be solved. Our scheme requires knowledge only on the local, not the global, lifetimes; (e) the method is an empirical method which does not involve any dynamical model, i.e. the forces which cause the circulation are not required. The method only finds that kinematic state of the atmosphere which, according to the continuity equation, fits best to the measurements. These kinematic state values are provided as model diagnostics to assess the performance of

10 dynamical models. Due to these advantages, the major problems in the empirical analysis of the Brewer-Dobson circulation as mentioned by Butchart (2014) are solved. Problems related to our method are (a) sensitivity of the inferred kinematic quantities



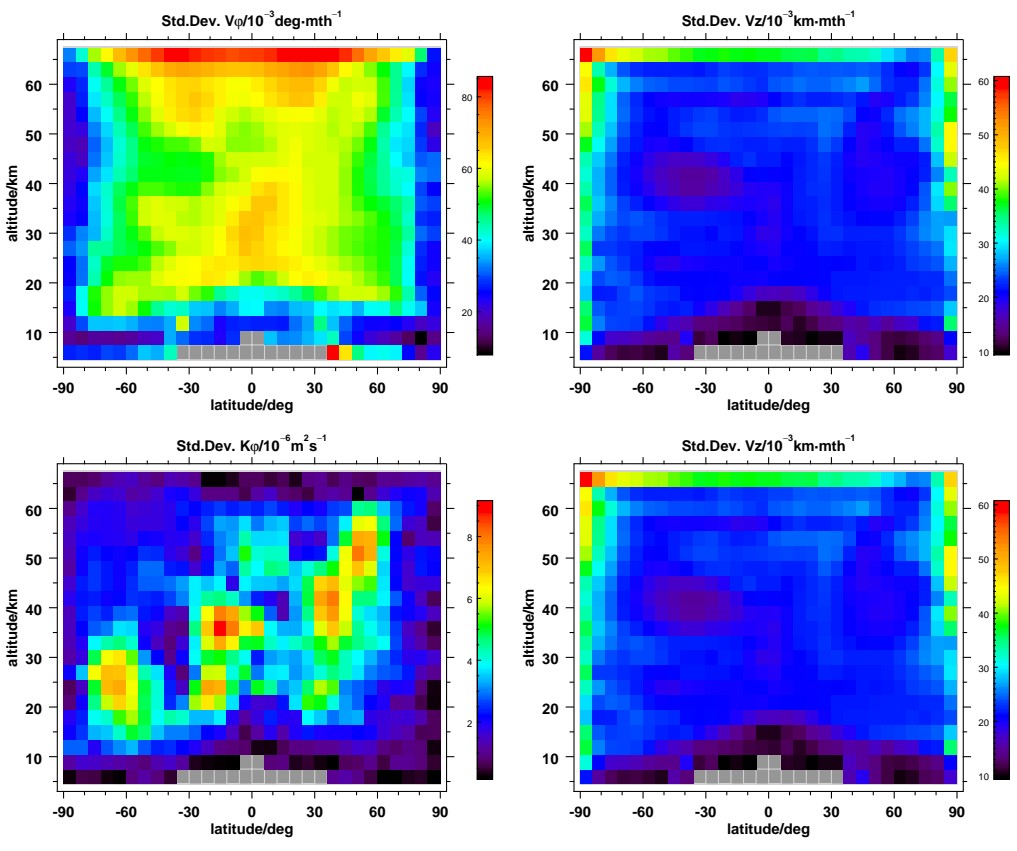

**Figure 3.** : Estimated uncertainties of $v_\phi(z, \phi)$, (upper left panel), $v_z(z, \phi)$ (upper right panel), $K_{phi}$ (lower left panel) and $K_z$ (lower right panel).

to locally varying biases, (b) a tendency towards ill-posedness of the inversion if distributions of too few tracers with too similar morphology are used, and (c) the usual artefacts arising if the numerical discretization is chosen too coarse. Results of the case study presented in Section 5.3 suggest that these problems have successfully been solved in the current application of the proposed scheme.

## 7   Conclusions and outlook

We have presented a method which infers mixing coefficients and effective velocities of a 2-D atmosphere by inversion of the continuity equation. The main steps of this procedure are (a) integration of the continuity equation over time to predict pressure and mixing ratios for given initial pressures and mixing ratios and initially guessed velocities and mixing coefficients; (b) propagation of errors of initial pressures and mixing ratios onto the predicted pressures and mixing ratios, by differentiation of





the predicted state with respect to the initial state and generalized Gaussian error propagation; (c) estimation of the sensitivities of the predicted state with respect to the velocities and mixing coefficients; and (d) minimization of a quadratic cost function involving the residual between measured and predicted state at the end of the forecasting interval by inversion of the continuity euqation. The inferred velocities are suggested to be used as a model diagnostic in order to avoid problems encountered with

other model diagnostics like mean age of stratospheric air. It is important to note that the diagnostics inferred here are effective transport velocities and effective mixing coefficients in a sense that they include eddy transport and diffusion terms. Thus, they cannot simply be compared to zonal mean velocities and mixing coefficients of a 3D model but the eddy terms have to be considered when these diagnostics quanities are calculated. The application of this method on $SF_6$ distributions measured by MIPAS (Stiller et al., 2012) to diagnose the Brewer-Dobson circulation are discussed in a companion paper. Obvious future

activities are the extension of this method to three dimensions and inclusion of sink functions of non-inert species to explore a larger number of tracers in order to better constrain the related inverse problem.

## Appendix A:  From 3D to 2D

The inference of effective two-dimensional transport velocities and effective mixing coefficients from measurements discussed in the main paper relies on the fact that under usual conditions (species chemical lifetimes large compared to transport lifetimes,

slow change of mean state compared to the effect of eddies) all eddy effects can be parametrized by transport and mixing terms. The additional velocities and mixing coefficients are, under the assumptions stated above, gas-independent. The formalism discussed below is largely based on Ko et al. (1985).

### A1   Eddy and mean flow transport

The change of partial number density ($\rho_g$) of a gas $g$ with time $t$ within a moving air parcel depends only on the net source

function. Diffusion, i.e. effects on scales not resolved by our system, are neglected:

$$\frac{\partial \rho_g}{\partial t} + \frac{\partial (u\,\rho_g)}{\partial x} + \frac{\partial (v\,\rho_g)}{\partial y} + \frac{\partial (w\,\rho_g)}{\partial z} = S_g \tag{A1}$$

The same reads in geographical coordinates, where $u$ and $v$ are redefined accordingly, using the shallow water approximation:

$$\frac{\partial \rho_g}{\partial t} + \frac{1}{r\cos\phi}\frac{\partial (u\,\rho_g)}{\partial \lambda} + \frac{1}{r}\frac{(\partial v\,\rho_g)}{\partial \phi} + \frac{\partial (w\,\rho_g)}{\partial z} = S_g \tag{A2}$$

Reynolds decomposition into zonal mean and eddy terms gives

$$\frac{\partial \rho_g}{\partial t} = S_g - \frac{1}{r\cos\phi}\frac{\partial((\overline{u}+u')(\overline{\rho_g}+\rho'_g))}{\partial \lambda} - \frac{1}{r}\frac{\partial((\overline{v}+v')(\overline{\rho_g}+\rho'_g))}{\partial \phi} - \frac{\partial((\overline{w}+w')(\overline{\rho_g}+\rho'_g))}{\partial z} \tag{A3}$$

Zonal averaging gives:

$$\frac{\partial \overline{\rho_g}}{\partial t} = \overline{S_g} - \overline{\frac{1}{r\cos\phi}\frac{\partial((\overline{u}+u')(\overline{\rho_g}+\rho'_g))}{\partial \lambda}} - \overline{\frac{1}{r}\frac{\partial((\overline{v}+v')(\overline{\rho_g}+\rho'_g))}{\partial \phi}} - \overline{\frac{\partial((\overline{w}+w')(\overline{\rho_g}+\rho'_g))}{\partial z}} \tag{A4}$$

Zonal mean longitudinal advection is zero.

$$\frac{\partial \overline{\rho_g}}{\partial t} = \overline{S_g} - \overline{\frac{1}{r}\frac{\partial((\overline{v}+v')(\overline{\rho_g}+\rho'_g))}{\partial \phi}} - \overline{\frac{\partial((\overline{w}+w')(\overline{\rho_g}+\rho'_g))}{\partial z}} \tag{A5}$$





Expansion gives

$$
\begin{aligned}
\frac{\partial \overline{\rho_g}}{\partial t} &= \overline{S_g} - \frac{1}{r}\frac{\partial(\overline{\overline{v}\,\rho_g + \overline{v}\,\rho_g' + v'\,\overline{\rho_g} + v'\,\rho_g'})}{\partial \phi} - \frac{\partial(\overline{\overline{w}\,\rho_g + \overline{w}\,\rho_g' + w'\,\overline{\rho_g} + w'\,\rho_g'})}{\partial z} \\
&= \overline{S_g} - \frac{1}{r}\frac{\partial(\overline{\overline{v}\,\rho_g} + \overline{\overline{v}\,\rho_g'} + \overline{v'\,\overline{\rho_g}} + \overline{v'\,\rho_g'})}{\partial \phi} - \frac{\partial(\overline{\overline{w}\,\rho_g} + \overline{\overline{w}\,\rho_g'} + \overline{w'\,\overline{\rho_g}} + \overline{w'\,\rho_g'})}{\partial z}.
\end{aligned}
\tag{A6}
$$

Zonal averages are constant and thus can be factorized. Zonal averages of linear functions of zonal perturbations are by definition zero, and we get

$$
\frac{\partial \overline{\rho_g}}{\partial t} = \overline{S_g} - \frac{1}{r}\frac{\partial(\overline{v}\,\overline{\rho_g} + \overline{v'\,\rho_g'})}{\partial \phi} - \frac{\partial(\overline{w}\,\overline{\rho_g} + \overline{w'\,\rho_g'})}{\partial z}.
\tag{A7}
$$

This can be rewritten in terms of volume mixing ratios, if number-density-weighted zonal averaging is performed.

$$
\frac{\partial \overline{vmr_g}}{\partial t} = \frac{\overline{S_g}}{\overline{\rho}} - \frac{1}{r}\frac{\partial(\overline{v}\,\overline{vmr_g} + \overline{v'\,vmr_g'})}{\partial \phi} - \frac{\partial(\overline{w}\,\overline{vmr_g} + \overline{w'\,vmr_g'})}{\partial z},
\tag{A8}
$$

where $\rho$ is air density.

The only term which causes that zonal mean transport is different from transport described by zonal mean velocity and mixing ratio gradient are the eddy flux terms $\frac{1}{r}\overline{v'vmr_g'}$ and $\overline{w'\,vmr_g'}$. In the following section these terms shall be further investigated.

## A2  Estimation of mixing ratio perturbances

To quantitatively asses the eddy terms, an estimate of the mixing ratio perturbations is needed. For this purpose, Equation (A8) is further simplified by the assumption that the velocity field is non-divergent:

$$
\frac{\partial \overline{vmr_g}}{\partial t} = \frac{\overline{S_g}}{\overline{\rho}} - \frac{\overline{v}}{r}\frac{\partial \overline{vmr_g}}{\partial \phi} - \frac{\partial}{\partial \phi}\overline{v'\,vmr_g'} - \overline{w}\frac{\partial \overline{vmr_g}}{\partial z} - \frac{\partial}{\partial z}\overline{w'\,vmr_g'}
\tag{A9}
$$

We divide Equation (A2) by the zonal mean air density and subtract Equation (A9) to get

$$
\begin{aligned}
\frac{\partial vmr_g}{\partial t} - \frac{\partial \overline{vmr_g}}{\partial t} + \frac{1}{r\cos\phi}\frac{\partial(u\,vmr_g)}{\partial \lambda} + \frac{1}{r}\frac{(\partial v\,vmr_g)}{\partial \phi} + \frac{\partial(w\,vmr_g)}{\partial z} = \\
S_g - \overline{S_g} + \frac{\overline{v}}{r}\frac{\partial \overline{vmr_g}}{\partial \phi} + \frac{\partial}{\partial \phi}\overline{v'\,vmr_g'} + \overline{w}\frac{\partial \overline{vmr_g}}{\partial z} + \frac{\partial}{\partial z}\overline{w'\,vmr_g'}.
\end{aligned}
\tag{A10}
$$

Rearrangement and application of the definition of the perturbation of the source and mixing ratio terms gives

$$
\begin{aligned}
\frac{\partial vmr_g'}{\partial t} + \frac{1}{r\cos\phi}\frac{\partial(u\,vmr_g)}{\partial \lambda} + \frac{1}{r}\frac{(\partial v\,vmr_g)}{\partial \phi} + \frac{\partial(w\,vmr_g)}{\partial z} - \frac{\overline{v}}{r}\frac{\partial \overline{vmr_g}}{\partial \phi} - \frac{\partial}{\partial \phi}\overline{v'\,vmr_g'} - \overline{w}\frac{\partial \overline{vmr_g}}{\partial z} \\
- \frac{\partial}{\partial z}\overline{w'\,vmr_g'} = S_g'.
\end{aligned}
\tag{A11}
$$

Ignoring quadratic terms in the perturbation according to perturbation theory this simplifies to

$$
\frac{\partial vmr_g'}{\partial t} + \frac{1}{r\cos\phi}\frac{\partial(u\,vmr_g)}{\partial \lambda} + \frac{1}{r}\frac{(\partial v\,vmr_g)}{\partial \phi} + \frac{\partial(w\,vmr_g)}{\partial z} - \frac{\overline{v}}{r}\frac{\partial \overline{vmr_g}}{\partial \phi} - \overline{w}\frac{\partial \overline{vmr_g}}{\partial z} = S_g'.
\tag{A12}
$$

We further ignore meridional and vertical mean advection, which are much smaller than zonal advection and get

$$
\frac{\partial vmr_g'}{\partial t} + \frac{1}{r\cos\phi}\frac{\partial(u\,vmr_g)}{\partial \lambda} + \frac{1}{r}\frac{(\partial v'\,vmr_g)}{\partial \phi} + \frac{\partial(w'\,vmr_g)}{\partial z} = S_g'.
\tag{A13}
$$





We apply the assumption of divergence-free velocities also to the remaining terms:

$$\frac{\partial vmr'_g}{\partial t} + \frac{u}{r\cos\phi}\frac{\partial(vmr_g)}{\partial\lambda} + \frac{v'}{r}\frac{(\partial vmr_g)}{\partial\phi} + w'\frac{\partial(vmr_g)}{\partial z} = S'_g. \tag{A14}$$

Rearrangement and Reynolds decomposition of $u$ and $vmr_g$ gives

$$\left(\frac{\partial}{\partial t} + \frac{\overline{u}+u'}{r\cos\phi}\frac{\partial}{\partial\lambda}\right)vmr'_g + \frac{v'}{r}\frac{(\partial vmr_g)}{\partial\phi} + w'\frac{\partial(vmr_g)}{\partial z} = S'_g, \tag{A15}$$

where we have used that $\frac{\partial\overline{vmr_g}}{\partial\lambda} = 0$. Perturbation theory also applies to higher order terms involving $u'$ and $vmr'_g$:

$$\left(\frac{\partial}{\partial t} + \frac{\overline{u}}{r\cos\phi}\frac{\partial}{\partial\lambda}\right)vmr'_g + \frac{v'}{r}\frac{\partial\overline{vmr_g}}{\partial\phi} + w'\frac{\partial\overline{vmr_g}}{\partial z} = S'_g. \tag{A16}$$

Assuming that eddy time scales are much shorter than mean transport time scales, Equation (A15) can be solved to give $vmr'_g$. We introduce eddy displacements $\alpha'$ and $\beta'$ by implicit definitions

$$\left(\frac{\partial}{\partial t} + \frac{\overline{u}}{r\cos\phi}\frac{\partial}{\partial\lambda}\right)\alpha' = v' \tag{A17}$$

and

$$\left(\frac{\partial}{\partial t} + \frac{\overline{u}}{r\cos\phi}\frac{\partial}{\partial\lambda}\right)\beta' = w' \tag{A18}$$

These equations state that the substantial change of the latitudinal displacement equals the latitudinal velocity perturbation, and mutatis mutandum for vertical displacement and speed. This relies on the assumption that the zonal mean mixing ratio changes much slower than the mixing ratio perturbations.

Similarly we get for the eddy chemical term $S'_g$

$$\left(\frac{\partial}{\partial t} + \frac{\overline{u}}{r\cos\phi}\frac{\partial}{\partial\lambda}\right)\gamma' = S'_g \tag{A19}$$

With these expressions the $v'$, $w'$ and $S'_g$ terms in Equation (A16) are replaced:

$$\left(\frac{\partial}{\partial t} + \frac{\overline{u}}{r\cos\phi}\frac{\partial}{\partial\lambda}\right)vmr'_g + \left(\frac{\partial}{\partial t} + \frac{\overline{u}}{r\cos\phi}\frac{\partial}{\partial\lambda}\right)\frac{\alpha'}{r}\frac{\partial\overline{vmr_g}}{\partial\phi} + \left(\frac{\partial}{\partial t} + \frac{\overline{u}}{r\cos\phi}\frac{\partial}{\partial\lambda}\right)\beta'\frac{\partial\overline{vmr_g}}{\partial z} = \tag{A20}$$
$$\left(\frac{\partial}{\partial t} + \frac{\overline{u}}{r\cos\phi}\frac{\partial}{\partial\lambda}\right)\gamma'$$

This allows to calculate $vmr'_g$:

$$vmr'_g = -\frac{\alpha'}{r}\frac{\partial\overline{vmr_g}}{\partial\phi} - \beta'\frac{\partial\overline{vmr_g}}{\partial z} + \gamma' \tag{A21}$$

**A3  Evaluation of the eddy flux**

The eddy flux terms can now be written as

$$\frac{1}{r}\overline{v'vmr'_g} = \overline{\frac{v'}{r}\left(-\frac{\alpha'}{r}\frac{\partial\overline{vmr_g}}{\partial\phi} - \beta'\frac{\partial\overline{vmr_g}}{\partial z} + \gamma'\right)} = \tag{A22}$$
$$-\frac{1}{r^2}\overline{v'\alpha'}\frac{\partial\overline{vmr_g}}{\partial\phi} - \frac{\overline{v'\beta'}}{r}\frac{\partial\overline{vmr_g}}{\partial z} + \frac{1}{r}\overline{v'\gamma'} =$$
$$-\frac{1}{r^2}K_{\phi\phi}\frac{\partial\overline{vmr_g}}{\partial\phi} - \frac{1}{r}K_{\phi z}\frac{\partial\overline{vmr_g}}{\partial z} + \frac{1}{r}\overline{v'\gamma'}$$





and

$$\overline{w'vmr'_g} = \overline{w'\left(-\frac{\alpha'}{r}\frac{\partial \overline{vmr_g}}{\partial \phi} - \beta'\frac{\partial \overline{vmr_g}}{\partial z} + \gamma'\right)} = \tag{A23}$$

$$-\frac{1}{r}\overline{w'\alpha'}\frac{\partial \overline{vmr_g}}{\partial \phi} - \overline{w'\beta'}\frac{\partial \overline{vmr_g}}{\partial z} + \overline{w'\gamma'} =$$

$$-\frac{1}{r}K_{z\phi}\frac{\partial \overline{vmr_g}}{\partial \phi} - K_{zz}\frac{\partial \overline{vmr_g}}{\partial z} + \overline{w'\gamma'}$$

where the following definitions have been used:

$$K_{\phi\phi} = \overline{v'\alpha'} \tag{A24}$$

$$K_{\phi z} = \overline{v'\beta'} \tag{A25}$$

$$K_{zz} = \overline{w'\beta'} \tag{A26}$$

$$K_{z\phi} = \overline{v'\alpha'} \tag{A27}$$

Equations (A22) and (A23) can be written in matrix notation:

$$\begin{pmatrix} \overline{w'vmr'_g} \\ \overline{v'vmr'_g} \end{pmatrix} = -\begin{pmatrix} K_{zz} & K_{z\phi} \\ K_{\phi z} & K_{\phi\phi} \end{pmatrix} \begin{pmatrix} \frac{\partial \overline{vmr_g}}{\partial z} \\ \frac{1}{r}\frac{\partial \overline{vmr_g}}{\partial \phi} \end{pmatrix} + \begin{pmatrix} \overline{w'\gamma'} \\ \overline{v'\gamma'} \end{pmatrix} \tag{A28}$$

For long-lived tracers, whose transport lifetimes are much longer than their chemical lifetimes, the chemical eddy term can be ignored and we get

$$\begin{pmatrix} \overline{w'vmr'_g} \\ \overline{v'vmr'_g} \end{pmatrix} = -\begin{pmatrix} K_{zz} & K_{z\phi} \\ K_{\phi z} & K_{\phi\phi} \end{pmatrix} \begin{pmatrix} \frac{\partial \overline{vmr_g}}{\partial z} \\ \frac{1}{r}\frac{\partial \overline{vmr_g}}{\partial \phi} \end{pmatrix} \tag{A29}$$

There is no explicit dependence of the eddy flux on the species. Thus, the same eddy flux tensor can be applied to all species, subject to the approximations used in itsderivation.

**A4   Analysis of the eddy tensor**

The eddy flux tensor can be decomposed into a symmetric and an antisymmetric part:

$$\begin{pmatrix} K_{zz} & K_{z\phi} \\ K_{\phi z} & K_{\phi\phi} \end{pmatrix} = \begin{pmatrix} K_{zz} & K^*_{z\phi} \\ K^*_{\phi z} & K_{\phi\phi} \end{pmatrix} + \begin{pmatrix} 0 & \psi \\ -\psi & 0 \end{pmatrix} \tag{A30}$$

where the following definitions are used:

$$K^*_{\phi z} = K^*_{z\phi} = \frac{K_{z\phi} + K_{\phi z}}{2} \tag{A31}$$

and

$$\psi = \frac{K_{z\phi} - K_{\phi z}}{2} \tag{A32}$$





### A4.1 The symmetric part

The fraction of the eddy flux governed by the symmetric part of the eddy flux tensor is

$$
\left( \frac{\partial \overline{vmr_g}}{\partial t} \right)_{symm} =
\tag{A33}
$$

$$
-\frac{1}{r}\frac{\partial}{\partial \phi}\left(\overline{v'vmr'}\right)_{symm} - \frac{\partial}{\partial z}\left(\overline{w'vmr'}\right)_{symm} =
$$

$$
-\nabla \cdot \left( \begin{array}{c} \overline{w'vmr'} \\ \overline{v'vmr'} \end{array} \right)_{symm} =
$$

$$
-\nabla \cdot \left( \left( \begin{array}{cc} K_{zz} & K_{z\phi}^* \\ K_{\phi z}^* & K_{\phi\phi} \end{array} \right) \left( \begin{array}{c} \frac{\partial \overline{vmr_g}}{\partial z} \\ \frac{1}{r}\frac{\partial \overline{vmr_g}}{\partial \phi} \end{array} \right) \right),
$$

where the shallow water approximation is used in the third line to get the form of a divergence. Symmetric tensors can be diagonalized:

$$
\mathbf{Q}^{-1} \left( \begin{array}{cc} K_{zz} & K_{z\phi}^* \\ K_{\phi z}^* & K_{\phi\phi} \end{array} \right) \mathbf{Q} = \left( \begin{array}{cc} K_{1,1} & 0 \\ 0 & K_{2,2} \end{array} \right)
\tag{A34}
$$

Using

$$
\mathbf{Q}^{-1} = \left( \begin{array}{cc} q_{1,1} & q_{1,2} \\ q_{2,1} & q_{2,2} \end{array} \right)^{-1} = \left( \begin{array}{cc} r_{1,1} & r_{1,2} \\ r_{2,1} & r_{2,2} \end{array} \right) = \mathbf{R}
\tag{A35}
$$

With $\mathbf{QR}$ being unity, Equation (A33) can be rewritten as

$$
\left( \frac{\partial \overline{vmr_g}}{\partial t} \right)_{symm} =
\tag{A36}
$$

$$
-\nabla \cdot \left( \mathbf{QR} \left( \begin{array}{cc} K_{zz} & K_{z\phi}^* \\ K_{\phi z}^* & K_{\phi\phi} \end{array} \right) \mathbf{QR} \left( \begin{array}{c} \frac{\partial \overline{vmr_g}}{\partial z} \\ \frac{1}{r}\frac{\partial \overline{vmr_g}}{\partial \phi} \end{array} \right) \right) =
$$

$$
-\nabla \cdot \left( \mathbf{Q} \left( \begin{array}{cc} K_{1,1} & 0 \\ 0 & K_{2,2} \end{array} \right) \mathbf{R} \left( \begin{array}{c} \frac{\partial \overline{vmr_g}}{\partial z} \\ \frac{1}{r}\frac{\partial \overline{vmr_g}}{\partial \phi} \end{array} \right) \right).
$$

This can be written component-wise:

$$
\left( \frac{\partial \overline{vmr_g}}{\partial t} \right)_{symm} = -\frac{1}{r^2}\frac{\partial}{\partial \phi} q_{2,2} K_{2,2} r_{2,2} \frac{\partial \overline{vmr_g}}{\partial \phi} - \frac{\partial}{\partial z} q_{1,1} K_{1,1} r_{1,1} \frac{\partial \overline{vmr_g}}{\partial z}
\tag{A37}
$$

This expression has the formal structure of a Fickian diffusion equation. Thus that part of the Eddy flux which is associated with the symmetric part of the eddy flux tensor can be understood as an additional mixing term.





### A4.2 The antisymmetric part

The fraction of the eddy flux governed by the antisymmetric part of the eddy flux tensor is

$$
\begin{aligned}
\left(\frac{\partial \overline{vmr_g}}{\partial t}\right)_{anti} &= -\frac{1}{r}\frac{\partial}{\partial \phi}\left(\overline{v'vmr'}\right)_{anti} - \frac{\partial}{\partial z}\left(\overline{w'vmr'}\right)_{anti} \\
&= -\nabla \cdot \left(\begin{pmatrix} 0 & \psi \\ -\psi & 0 \end{pmatrix}\begin{pmatrix} \frac{\partial \overline{vmr_g}}{\partial z} \\ \frac{1}{r}\frac{\partial \overline{vmr_g}}{\partial \phi} \end{pmatrix}\right) \\
&= -\frac{1}{r}\frac{\partial}{\partial \phi}\left(\psi\frac{\partial \overline{vmr_g}}{\partial z}\right) + \frac{\partial}{\partial z}\left(\psi\frac{1}{r}\frac{\partial \overline{vmr_g}}{\partial \phi}\right) \\
&= -\frac{1}{r}\frac{\partial \psi}{\partial \phi}\frac{\partial \overline{vmr_g}}{\partial z} - \frac{\psi}{r}\frac{\partial}{\partial \phi}\frac{\partial \overline{vmr_g}}{\partial z} + \frac{1}{r}\frac{\partial \psi}{\partial z}\frac{\partial \overline{vmr_g}}{\partial \phi} - \frac{\psi}{r^2}\frac{\partial \overline{vmr_g}}{\partial \phi} + \frac{\psi}{r}\frac{\partial}{\partial z}\frac{\partial \overline{vmr_g}}{\partial \phi} \\
&= -\frac{1}{r}\frac{\partial \psi}{\partial \phi}\frac{\partial \overline{vmr_g}}{\partial z} + \left(\frac{\partial \psi}{\partial z} - \frac{\psi}{r}\right)\frac{1}{r}\frac{\partial \overline{vmr_g}}{\partial \phi}
\end{aligned}
\tag{A38}
$$

We introduce virtual velocities

$$
\tilde{v} = \frac{\partial \psi}{\partial z} - \frac{\psi}{r}
\tag{A39}
$$

and

$$
\tilde{w} = \frac{1}{r}\frac{\partial \psi}{\partial \phi};
\tag{A40}
$$

With this we get

$$
\left(\frac{\partial \overline{vmr_g}}{\partial t}\right)_{anti} = -\tilde{v}\frac{1}{r}\frac{\partial \overline{vmr_g}}{\partial \phi} - \tilde{w}\frac{\partial \overline{vmr_g}}{\partial z}
\tag{A41}
$$

This expression has the formal structure of a transport equation. Thus, that part of the Eddy flux which is associated with the antisymmetric part of the eddy flux tensor can be understood as an additional transport term.

*Acknowledgements.* TvC wishes to thank Hendrik Elbern, Richard Menard, Peter Braesicke, Björn-Martin Sinnhuber and Thomas Birner for drawing his attention to some important literature and for encouragement.



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
