# Peer review of "Direct Inversion of Circulation and Mixing from Tracer Measurements: I. Method"

_Atmospheric Chemistry and Physics, 2016_

## Referee Comment (RC1) · Anonymous Referee #2 · 20 Jul 2016

**General comment:**

The paper presents a method for deducing atmospheric circulation (wind field) and mixing parameters from trace gas measurements by inversion of the continuity equation. In a first step, the mathematical framework is defined and explained. Second, the method is applied to idealized tracer fields and to MIPAS satellite measurements (in the "proof of concept" section 5), to show that the inversion indeed results in reliable velocities and diffusivities.

Deducing information about the circulation from measurements, without involving information from models, is a great challenge in atmospheric sciences. This paper seems to contain an important contribution to reach that goal, what renders it definitely publishable and of great interest to a large readership of ACP. However, I have two major points which the authors need to assess before publication. First, the paper is not easy-to-read and the presentation quality needs improvement - otherwise I feel that the paper will fail in addressing a large readership. Second, I have some concerns about the so-called "proof of concept".

**Major comments:**

1) Presentation and Notation:

Overall, the paper is overloaded with detailed formulae, but lacking motivating and explanatory paragraphs. In their own words (P2, L33), the authors aim to avoid "that the reader does not see the forest for the trees" but, in my opinion, there are still too many trees around. For instance in section 3, there should be a clear motivation at the beginning, why the derivatives (which are calculated in the following) are needed and what the matrix notation means. After that the equations (15-26) could be nicely combined into one single equation-array (similarly in section 4, starting with equation (37)). Concerning all formulae, writing $X, \chi, \mu, ...$ for mixing ratio instead of $vmr$ would help to increase readability.

Moreover, while many steps in the calculation are written in detail (like taking derivatives), at some points I was not able to understand the derivations in detail. One such example is the matrix notation in equation (27). First, a clear motivation should be given why this matrix notation is advatageous and what it means (this is the heart of the paper). Second, I did not succeed in understanding the dimensionalities of the quantities involved. As the authors state, the D-matrix is build from three submatrices of dimensions $K_0 \times K_0$ ($I_K$), $K_0 \times 2K_0$ ($W_i$), and $J_0 \times L_0$ ($D_{\rho,nom}$). Therefore, the D-matrix has dimensions $(2K_0 + J_0) \times (3K_0 + L_0)$, which is, as far as I can see, not consistent with the vector it is acting on. Please check the dimensionalities again and explain clearly what equation (27) means. Equation (35) caused me similar problems with understanding. Please explain clearly where it comes from.
The appendix is, in my opinion, not necessary. It just presents a recalculation of the existing literature. I would recommend to reduce such recalculations, but to add explanations at the critical and new steps of this paper (e.g., around Eqns. 27/35). If the authors want to keep that part, it could be moved to the supplementary material.

2) "Proof of concept":

In my opinion, section 5.3 does not really present a "proof of concept", as promised by the title of section 5. The method is used to deduce velocities and diffusivities from tracer measurements, but the true underlying circulation is not known. Therefore, this case is no proof that the inversion method yields the correct result. I think, for a true "proof of concept" the circulation and diffusivities must be known before and need to be reproduced by the method. Section 5.2 points into this direction, but is exclusively descriptive. The optimal "proof of concept" would be to have a 2D-model based on equations (3-4) with idealized velocities and diffusivities, and to invert the resulting trace gas distributions. At least, the cases described in section 5.2 should be explained in more detail and related results should be shown in the paper.

**Specific comments:**

P2, L1ff: Another source of uncertainty when deducing mean age from $SF_6$ is related to the fact that the tropospheric increase is not strictly linear (see Garcia et al., 2011, JAS).

P2, L7: To my knowledge, in models usually the surface layer is used as a reference, not the upper edge of the TTL. Please clarify.

P18, L11: How robust are the deduced velocities and diffusivities with respect to the choice of initial value for the iteration. Please give some quantitative estimate.

P20, L7: How can the residual be small for $SF_6$ if no chemical sink is included in the calculation? Is the sink effect absorbed in the transport terms, or is a significant sink only existing above the upper boundary for the calculation?

P20, L13: "Velocities are roughly consistent with mean ages...". Some misinterprations in the past arose from relating mean age simply to the stratospheric circulation. However, mean age is known to be controlled by circulation and mixing (e.g., Neu and Plumb, 1999; Garny et al., 2014; Ploeger et al., 2015). So please discuss carefully what you mean here with "consistent".

**Technical corrections:**

Equation (11): I think there should be no minus here (in the supplement there is also no minus: Eq. (34)).

Equation (13): Brackets missing around the argument of volume mixing ratio.

Equation (18): Missing point behind equation.

Equation (27): Are the dimensions correct - see my major comment 1.

P12, L15: Point behind "numerical artefacts."

P13, L6: K's should have an index $j$, like v and w.

P13, L15: vmr in italics - or better: use some symbol instead (e.g., X, see also my major commet 1).

P18, L27: "set to zero."

P19, L13: Replace "In order to fight..." by "Due to..."

P19, L27: "...macro timestep"

P20, L21: I guess you mean Figure 2.

P20, L23: Figure 3 has no middle column.

Figure 1: The figure, and particularly the descriptions need to be enlarged.

Figure 2: Give a color bar for the velocities in the upper/left panel. Caption: "...(upper right panel). Bottom panels show...". And write $K_\phi$ instead of $K_{phi}$.

Figure 3: The lower/right panel is the same as the upper/right - it should show K$_z$. Caption: Use $v, w$ instead of $v_\phi, ...$ and $K_\phi$, to be consistent with the rest of the paper.

**References:**

Garcia et al. (2011), J. Atmos. Sci., 68.

Garny et al. (2014), J. Geophys. Res., 119.

Neu and Plumb (1999), J. Geophys. Res., 104.

Ploeger et al. (2015), J. Geophys. Res., 120.

―――――――――――――――――――――

---

## Author Comment (AC1) · 22 Jul 2016

We thank the reviewer for the insightful and thorough comments which certainly will improve the paper. In the following we have included the reviewer's comments in *italic face*. Our replies are printed in normal face.

*General comment:*

*The paper presents a method for deducing atmospheric circulation (wind field) and mixing parameters from trace gas measurements by inversion of the continuity equation. In a first step, the mathematical framework is defined and explained. Second, the*

[Figure]

*method is applied to idealized tracer fields and to MIPAS satellite measurements (in the "proof of concept" section 5), to show that the inversion indeed results in reliable velocities and diffusivities.*

*Deducing information about the circulation from measurements, without involving information from models, is a great challenge in atmospheric sciences. This paper seems to contain an important contribution to reach that goal, what renders it definitely publishable and of great interest to a large readership of ACP. However, I have two major points which the authors need to assess before publication. First, the paper is not easy-to-read and the presentation quality needs improvement – otherwise I feel that the paper will fail in addressing a large readership. Second, I have some concerns about the so-called "proof of concept".*

We thank the reviewer for this encouraging general comment. We are confident that we will be able to improve the presentation quality. The specific comments of the reviewer are of great help here. With respect to the issue of the "proof of concept", please see our reply below.

*Major comments:*

*1) Presentation and Notation:*

*Overall, the paper is overloaded with detailed formulae, but lacking motivating and explanatory paragraphs. In their own words (P2, L33), the authors aim to avoid "that the reader does not see the forest for the trees" but, in my opinion, there are still too many trees around. For instance in section 3, there should be a clear motivation at the beginning, why the derivatives (which are calculated in the following) are needed and what the matrix notation means.*

We agree that some sentences of motivation of certain steps will be helpful and we will include some.

*After that the equations (15-26) could be nicely combined into one single equation-array (similarly in section 4, starting with equation (37)).*

The sentences between the equations are meant to guide the reader through the forest of equations and to motivate what each single equation is good for. We think that without these explaining sentences the general criticism that there are too many equations and too little explaining and motivating text would be even more applicable.

*Concerning all formulae, writing $X$; $x$; $\mu$; . . . for mixing ratio instead of vmr would help to increase readability.*

The problem with our paper is that it is interdisciplinary, addressing the communities of remote sensing, inverse theory, atmospheric modelling etc. The suggested notation would clash with the conventions in some of the communities mentioned. For example, $x$ is usually the independent variable in inverse theory, and $\mu$ is often used as expectation value. In our own context, we need variable $x$ later for the combined state vector. Thus, we are inclined to stay with the self-explaining variable name.

*Moreover, while many steps in the calculation are written in detail (like taking derivatives), at some points I was not able to understand the derivations in detail. One such example is the matrix notation in equation (27). First, a clear motivation should be given why this matrix notation is advantageous...*

[Figure]

The matrix notation has been chosen because with this the formalism can be expressed in a much more compact manner. We will include a note on this.

*and what it means (this is the heart of the paper).*

We agree that this is the heart of the paper and deficiencies in clarity at this point would be detrimental. On the other hand, the meaning of the terms is explained in the itemization after Eq. (27). We have decided for the following to make this part more transparent: We will include a figure which will illustrate which block of the D-matrix operates on which components of the initial state vector to produce which components of the final state vector. In the itemization after Eq. (27) we will then refer to this visualization. We are inclined to follow this approach because we think that a pure verbal description would make the paper more tedious to read, would add unnecessary length, and would not necessarily add clarity.

*Second, I did not succeed in understanding the dimensionalities of the quantities involved. As the authors state, the D-matrix is build from three submatrices of dimensions $K_0 \times K_0$ $(I_K)$, $K_0 \times 2K_0$ $(W_i)$, and $J_0 \times L_0$ $(D_{\rho;nom})$. Therefore, the D-matrix has dimensions $(2K_0 + J0) \times (3K_0 + L_0)$, which is, as far as I can see, not consistent with the vector it is acting on. Please check the dimensionalities again and explain clearly what equation (27) means.*

The D-matrix is not block-diagonal, that means, the dimensions of the sub-matrices it is formed of cannot simply be added to give the total dimension. Some of the sub-matrices act on the same components of the input state vector. As written above we will visualize this.

*Equation (35) caused me similar problems with understanding. Please explain clearly where it comes from.*

Here the same explanation holds as sketched above.

*The appendix is, in my opinion, not necessary. It just presents a recalculation of the existing literature. I would recommend to reduce such recalculations, but to add explanations at the critical and new steps of this paper (e.g., around Eqns. 27/35). If the authors want to keep that part, it could be moved to the supplementary material.*

What has been published as an appendix in the discussion paper was initially just an internal document to help ourselves to better understand the issue of eddy transport and why zonal averaging can cause additional transport and mixing components. It was initially not foreseen for publication. It was not part of the initial submission. The other reviewer, however, requested a more thorough discussion of these issues, and due to that, we have decided to include this. Due to the lack of original content, we have decided to include it as appendix only. To provide this as a supplement might indeed be a good compromise. **Since inclusion of this was triggered by a comment in the other access review, a recommendation on this issue by the other reviewer would be highly appreciated.**

*2) "Proof of concept":*

*In my opinion, section 5.3 does not really present a "proof of concept", as promised by the title of section 5. The method is used to deduce velocities and diffusivities from tracer measurements, but the true underlying circulation is not known. Therefore, this case is no proof that the inversion method yields the correct result.*

This is why in Section 5.1. and 5.2. we test the method using idealized cases where the functionality of the components can be easily judged.

*I think, for a true "proof of concept" the circulation and diffusivities must be known before and need to be reproduced by the method. Section 5.2 points into this direction, but is exclusively descriptive. The optimal "proof of concept" would be to have a 2D-model based on equations (3-4) with idealized velocities and diffusivities, and to invert the resulting trace gas distributions. At least, the cases described in section 5.2 should be explained in more detail and related results should be shown in the paper.*

We have chosen the title "proof of concept" in order to avoid the more ambitious term "validation". While validation aims at providing evidence that the system provides the correct results in a quantitative sense, the claim of a proof of concept is, to our understanding, much weaker. A "concept" is little more than a quite general idea, and a "proof of concept" just shall generate confidence that it is worthwhile to pursue these activities, that there is no evidence that the concept leads inavoidably into a blind alley etc. Actions towards validation of this method have been initiated, including comparisons with models etc, but these are far beyond the scope of this methodical paper.

In reply to this comment, we will take the following actions:

1. We will add some text where we describe what the proof of concept is meant to be, in order to avoid to raise false expectations.

2. We will present some of the studies undertaken under 5.2.

*Specific comments:*

*P2, L1ff: Another source of uncertainty when deducing mean age from $SF_6$ is related to the fact that the tropospheric increase is not strictly linear (see Garcia et al., 2011,*

[Figure]

*JAS).*

We agree that non-linearity of tropospheric $SF_6$ is an issue. However, this non-linearity in the tropospheric $SF_6$ time series is considered in the work of, e.g., Stiller et al (2012) or Haenel et al. (2015). These authors use an iterative scheme to infer the age. As first guess, they use the age directly inferred from the $SF_6$ mixing ratio. Then, they calculate the age spectrum for this initially guessed age, convolve the (non-linear) tropospheric time series with this age-dependent age spectrum, infer a correction, and iterate until convergence. Thus, the only remaining uncertainty in this context is the uncertainty of the age spectrum. However we agree that not every reader might be aware of this method, and we will add a note on the non-linearity issue.

*P2, L7: To my knowledge, in models usually the surface layer is used as a reference, not the upper edge of the TTL. Please clarify.*

There seems to be some disagreement in the modelling community. At least, in a review of a paper by Stiller et al., 2012, the reviewer insisted that the age of air is defined relative to the time of entry into the stratosphere, and the reviewer was not at all happy with our reply that an age defined like this is an empirically void theoretical quantity. But if the models indeed use surface as reference, then even better! Then their age indeed is a quantity with empirical content. We will reword the text accordingly.

*P18, L11: How robust are the deduced velocities and diffusivities with respect to the choice of initial value for the iteration. Please give some quantitative estimate.*

In a constrained retrieval, where the initial guess is set equal to the a priori field, there are generally two possible mechanisms for a dependence of the result on the initial

guess.

1. **The effect of the constraint:** The solution of a constrained inversion always has a tendency to be pulled towards the a priori field, or, in our case with 1st order Tikhonov constraint, the field gradients are constrained to those of the a priori field. Our a priori constraint is chosen as zero velocity throughout. Since the resulting field gradients deviate largely from this initial guess and resemble those of the expected velocity fields, it is evident that the inversion is able to find a solution which is far away from the initial guess and is not overly constrained by the a priori assumption.

2. **Non-linearity and possible secondary minima of the cost function:** The observed convergence rate indicates an almost linear inverse problem; thus, no such related problems are seen. We have experienced that problematic cases regularily end up in non-converging iterations rather than converging to different results and are thus easily sorted out.

Our results provide the smoothest field of velocities and the smallest mixing coefficients which are still consistent with the measurements. Any change of the a priori assumptions must have an impact on the results. How large this intentional dependence is, is fully user-controllable by adjustment of the regularisation strength.

*P20, L7: How can the residual be small for $SF_6$ if no chemical sink is included in the calculation? Is the sink effect absorbed in the transport terms, or is a significant sink only existing above the upper boundary for the calculation?*

The problem with the sink in Stiller et al. (2012) is, that age-calculations are sensitive to the accumulated decomposition since the air has left the troposphere. In our case, only the decomposition during the finite time-step of the calculation can contribute,

because the atmospheric state at the beginning of the time step already is depleted in the trace gases. For $SF_6$, the major sink is indeed above the altitude range under consideration.

*P20, L13: "Velocities are roughly consistent with mean ages...". Some misinterprations in the past arose from relating mean age simply to the stratospheric circulation. However, mean age is known to be controlled by circulation and mixing (e.g., Neu and Plumb, 1999; Garny et al., 2014; Ploeger et al., 2015). So please discuss carefully what you mean here with "consistent".*

We emphasize the attribute "roughly". This statement is not meant as a quantitative assessment but is meant to say that there is no obvious major contradiction between our results and our current knowledge on stratospheric mean age of air. We will change the text towards a more careful wording.

*Technical corrections:...*

We are grateful for the thorough reading of the manuscript and will carefully apply the corrections.

---

## Referee Comment (RC2) · Anonymous Referee #1 · 10 Aug 2016

This paper is clearly the result of a major and impressive undertaking with substantial investment by the authors – chapeau! The paper is centered around the numerical treatment of the inverse problem of deriving transport characteristics from tracer measurements, which by itself is novel and has the potential to make a fundamental and important contribution so should be published. My main concerns are with the physical interpretation of the inferred transport characteristics and the approximations used to derive the tracer continuity equations. As outlined in my major comment below, I think the authors need to include more discussion of these potential issues. I also have a few minor and editorial comments that should be taken into account before publication.

Major Comments:

[Figure]

The purpose of the approach is to apply it to zonal-mean atmospheric tracer data. The corresponding continuity and tracer continuity Eq's are supposedly those arising from the zonally averaged 3-d Eq's, but in fact they are not, according to the derivations in the appendix. On line 7, page 25 it is claimed that density-weighting is performed (as is in section 3.1, line 6 on page 5): this would require redefining the zonal average and, more importantly, redefining the eddy part of the Reynolds decomposition. Also in the appendix, on line 15 of page 25 it is stated that the velocity field is assumed to be non-divergent. This essentially corresponds to applying a Boussinesq approximation, which is what the referenced Ko et al. (1985) use (discussed in their appendix). But applying a Boussinesq approximation means that the (relevant) density perturbations are neglected and therefore no density-weighting is used. I am skeptical that a Boussinesq approximation is suitable for this problem, although it's possible that this is less of an issue in the height coordinates used here (it most certainly is an issue for the isentropic coordinates used by Ko et al).

I recommend consulting Tung's 1986 paper (J. Atmos. Sci., 43, pages 2600-2618) that lays out a zonal-mean framework for isentropic coordinates and includes a detailed description of the mathematical treatment of density-weighting (his section 2). Similar frameworks apply to other coordinate systems; an exception are pressure coordinates (or log-pressure coordinates) because they implicitly include mass/density-weighting.

I think these issues should be discussed in the appendix. I recommend focusing on the conceptual points (approximations needed to make eddy tensor independent of tracer; interpretation of eddy tensor and symmetric / antisymmetric components as additional mixing / mean advection); the detailed mathematical treatment is not necessary I think or at least can be significantly condensed.

Related to the above, I think the main description in section 3.1 needs more physical interpretation. The meanings of "effective transport velocity" and "eddy diffusion" are not unique, but depend on the kinds of approximations and treatment of perturbations outlined in the appendix. A different zonal mean / eddy treatment and different approximations would result in different effective transport velocities and eddy diffusivities. In fact, I would argue that the only unique definition of "effective transport velocity" would be one that sets the eddy diffusivities to zero (ad hoc), thereby absorbing all resolved transport into zonal-mean advection. After all, "eddy diffusion" is a parametrization of eddy advection and almost all transport is in reality advective (although, of course, some small-scale / sub-grid scale diffusion is still necessary to close the system). I am sure the authors have thought about the physical interpretation of their deduced transport contributions and I would like to see some of those thoughts discussed somewhere in the paper. The way it's currently written makes it sound too much like the inferred transport characteristics are unique.

Minor Comments, incl. editorial:

page 1, line 15: "are under debate" feels out-of-date: as suggested by the introduction and by the other reviewer comments, a lot of what seemed to be a debate at first has become a more detailed and nuanced description

page 1, line 21: "is not that one-dimensional": somewhat confusing what this refers to (I suppose it refers to the spatial variability in age trends - see following sentence in text)

page 1, line 24: I think you either need to spell out the "ad hoc assumptions" or remove this remark

page 2, line 6/7: I think most models in practice use the Earth's surface as a reference (see e.g. CCMVal reports)

page 3, line 11: suggest "source and sink terms" for clarity

page 4, line 2: "shallowness approximation": it will help some readers to spell out the approximation (i.e. z much smaller than r_E)

page 7, line 28: differentiate

page 10, line 4: I'd prefer "identity matrix"

page 12, line 4: equal to an integer multiple . . .

page 12, line 12: I suggest "real atmosphere"

page 19, line 22/23: velocities should be v and w, subscript for $K_\phi$ (also Fig. caption of Fig. 3)

page 20, line 13: "roughly consistent" should be specified/quantified more - in what way are they consistent (list a few examples, such as overall latitudinal and vertical structures etc)?

page 20, line 14/15: "shows many detailed features demanding scientific investigation in their own right": which detailed features and why to they demand investigation? I think this should be discussed in the text.

page 20, line 17: should be "mixing coefficients"

page 20, line 25: weekly

page 20, line 33: view _of_ the same problem

Fig. 1: right-hand column misses a color bar; labels are hard to read - please increase font size

Fig. 2 caption: "a detail of this" - I suggest "a zoomed-in view"
* * *

---

## Author Comment (AC2) · 7 Oct 2016

The reply is found in the supplement.

Please also note the supplement to this comment:
http://www.atmos-chem-phys-discuss.net/acp-2016-322/acp-2016-322-AC2-supplement.pdf

[Figure]

The authors thank the reviewers for their encouraging and helpful comments. In the following, the comments are printed in **bold face**, our replie in normal face, and the actions taken to improve the manuscript in *italic face*.

**Report 1**

Comment: **This paper is clearly the result of a major and impressive undertaking with substantial investment by the authors – chapeau! The paper is centered around the numerical treatment of the inverse problem of deriving transport characteristics from tracer measurements, which by itself is novel and has the potential to make a fundamental and important contribution so should be published. My main concerns are with the physical interpretation of the inferred transport characteristics and the approximations used to derive the tracer continuity equations. As outlined in my major comment below, I think the authors need to include more discussion of these potential issues. I also have a few minor and editorial comments that should be taken into account before publication.**

Reply: The authors thank the reviewer for this encouraging comment.

Action: *N/A.*

Comment: **Major Comments:**
**The purpose of the approach is to apply it to zonal-mean atmospheric tracer data. The corresponding continuity and tracer continuity Eqs are supposedly those arising from the zonally averaged 3-d Eqs, but in fact they are not, according to the derivations in the appendix. On line 7, page 25 it is claimed that density-weighting is performed (as is in section 3.1, line 6 on page 5): this would require redefining the zonal average and, more importantly, redefining the eddy part of the Reynolds decomposition. Also in the appendix, on line 15 of page 25 it is stated that the velocity field is assumed to be non-divergent. This essentially corresponds to applying a Boussinesq approximation, which is what the referenced Ko et al. (1985) use (discussed in their appendix). But applying a Boussinesq approximation means that the (relevant) density perturbations are neglected and therefore no density-weighting is used. I am skeptical that a Boussinesq approximation is suitable for this problem, although its possible that this is less of an issue in the height coordinates used here (it most certainly is an issue for the isentropic coordinates used by Ko et al). I recommend consulting Tungs 1986 paper (J. Atmos. Sci., 43, pages 2600-2618) that lays out a zonal-mean framework for isentropic coordinates and includes a detailed description of the mathematical treatment of density-weighting (his section 2). Similar frameworks apply to other coordinate systems; an exception**

**Fig. 1.**

**Supplement:**

The authors thank the reviewers for their encouraging and helpful comments. In the following, the comments are printed in **bold face**, our replie in normal face, and the actions taken to improve the manuscript in *italic face.*

**Report 1**

Comment: **This paper is clearly the result of a major and impressive undertaking with substantial investment by the authors – chapeau! The paper is centered around the numerical treatment of the inverse problem of deriving transport characteristics from tracer measurements, which by itself is novel and has the potential to make a fundamental and important contribution so should be published. My main concerns are with the physical interpretation of the inferred transport characteristics and the approximations used to derive the tracer continuity equations. As outlined in my major comment below, I think the authors need to include more discussion of these potential issues. I also have a few minor and editorial comments that should be taken into account before publication.**

Reply: The authors thank the reviewer for this encouraging comment.

Action: *N/A.*

Comment: **Major Comments:**
**The purpose of the approach is to apply it to zonal-mean atmospheric tracer data. The corresponding continuity and tracer continuity Eqs are supposedly those arising from the zonally averaged 3-d Eqs, but in fact they are not, according to the derivations in the appendix. On line 7, page 25 it is claimed that density-weighting is performed (as is in section 3.1, line 6 on page 5): this would require redefining the zonal average and, more importantly, redefining the eddy part of the Reynolds decomposition. Also in the appendix, on line 15 of page 25 it is stated that the velocity field is assumed to be non-divergent. This essentially corresponds to applying a Boussinesq approximation, which is what the referenced Ko et al. (1985) use (discussed in their appendix). But applying a Boussinesq approximation means that the (relevant) density perturbations are neglected and therefore no density-weighting is used. I am skeptical that a Boussinesq approximation is suitable for this problem, although its possible that this is less of an issue in the height coordinates used here (it most certainly is an issue for the isentropic coordinates used by Ko et al). I recommend consulting Tungs 1986 paper (J. Atmos. Sci., 43, pages 2600-2618) that lays out a zonal-mean framework for isentropic coordinates and includes a detailed description of the mathematical treatment of density-weighting (his section 2). Similar frameworks apply to other coordinate systems; an exception**

**are pressure coordinates (or log-pressure coordinates) because they implicitly include mass/density-weighting. I think these issues should be discussed in the appendix. I recommend focusing on the conceptual points (approximations needed to make eddy tensor independent of tracer; interpretation of eddy tensor and symmetric / antisymmetric components as additional mixing / mean advection); the detailed mathematical treatment is not necessary I think or at least can be significantly condensed.**

Reply: We agree that this part has been the week point in the initial submission. Since for us the correctness of the paper is more important than rapid publication, we would appreciate if the reviewer would have another critical look at the revised version.

Our inverse model will produce "effective velocities" and "effective mixing coefficients" instead of zonally averaged velocities and mixing coefficients, because of the eddy terms. The point is that the interpretation of these "effective kinematic quantities" depends on this choice and on the assumed approximations. The interpretation tells us which effects beyond the true zonal mean velocities and the true subscale effects are included in the "effective kinematic quantities". We have tackled the problem as follows:

First, we have studied the literature about 2D models involving mass-weighted zonal averages (Gallimore and Johnson, 1981, JAS **38**, 583-599 and 1870-1890; Tung, 1986, JAS **43**, 2600-2618). We found that the theory of mass-weighted zonal averaging is applied to $\theta$ coordinates in order to distinguish different dynamical processes. We are currently interested in the kinematic understanding and do not yet aim at a dynamical understanding (i.e. we want to know what the effective kinematic quantities are, and what they represent, but not by which forces they are caused). Further, our model is formulated in geometrical vertical coordinates. Thus, we have not pursued this path any further. We now work again with unweighted zonal averaged VMRs.

Second, we have studied the Tung 1986 paper, which left us somewhat clueless but contains a reference to Tung (1982, JAS 39 2330-2355) which appears helpful. In particular his formalism does not use the Boussinesq approximation about which we agree that it is not adequate for our purpose. We do not use it any more. In Tung (1982), a theory of a 2D model (non-weighted zonal means) is outlined which seems well applicable also to our case. We have taken the following steps:

1. we have checked that the formalism is also applicable to geometric vertical coordinates and have verified that the same approximations are valid.
2. we list the approximations made in this paper.
3. we have developed an interpretation of our kinematic quantities, which is based on these approximations.
4. we explicitly state that the interpretation of the inferred kinematic quantities depends on the assumptions and approximations made.

Action: *The entire appendix has been rewritten, based on Tung (1982). The new appendix is much shorter because we do not report the entire formalism, starting*

*from the 3D continuity equation, but we start from Eq. D14 in Tung (1982). We now present only those equations which are indispensable to understand our interpretation of the kinematic variables.*

*Comment:* **Related to the above, I think the main description in section 3.1 needs more physical interpretation. The meanings of "effective transport velocity" and "eddy diffusion" are not unique, but depend on the kinds of approximations and treatment of perturbations outlined in the appendix. A different zonal mean / eddy treatment and different approximations would result in different effective transport velocities and eddy diffusivities. In fact, I would argue that the only unique definition of "effective transport velocity" would be one that sets the eddy diffusivities to zero (ad hoc), thereby absorbing all resolved transport into zonal-mean advection. After all, "eddy diffusion" is a parametrization of eddy advection and almost all transport is in reality advective (although, of course, some small-scale / subgrid scale diffusion is still necessary to close the system). I am sure the authors have thought about the physical interpretation of their deduced transport contributions and I would like to see some of those thoughts discussed somewhere in the paper. The way its currently written makes it sound too much like the inferred transport characteristics are unique.**

*Reply: We agree.*

*Action: The assumptions, approximations, and interpretation of the involved quantities are now better specified. A clear statement is made that the interpretation of our effective kinematic quantities is not unique but depends on the approximations and assumptions made. As an alternative, it is now also proposed to use our method to validate 2D models, including their inherent assumptions and approximations to solve the eddy transport problem.*

*Comment:* **Minor Comments, incl. editorial:**
**page 1, line 15: "are under debate" feels out-of-date: as suggested by the introduction and by the other reviewer comments, a lot of what seemed to be a debate at first has become a more detailed and nuanced description**

*Reply: agreed*

*Action:Changed as follows:*
*OLD: are under debate*
*NEW: have become an important research topic*

*Comment:* **page 1, line 21: "is not that one-dimensional": somewhat confusing what this refers to (I suppose it refers to the spatial vari-**

**ability in age trends - see following sentence in text)**

*Reply: agreed*

*Action:Changed as follows:*
*OLD: ... not as one-dimensional. Instead, stratospheric age trends ...*
*NEW: ... more complex. Stratospheric age trends ...*

*Comment:* **page 1, line 24: I think you either need to spell out the "ad hoc assumptions" or remove this remark**

*Reply: agreed*

*Action:added: These include the adequacy of the Wald (inverse Gaussian) function for the representation of the age spectrum and the choice of its width parameter.*

*Comment:* **page 2, line 6/7: I think most models in practice use the Earth's surface as a reference (see e.g. CCMVal reports)**

*Reply: But not all. E.g. In Stiller et al. (2012, ACP **12** 3311-3331) a reviewer tried to force us to use the stratospheric entry point as reference, claiming that this was the usual reference.*

*Action:Reworded as follows:*
*OLD: ... the modelling community has established the upper edge of the tropical tropopause layer as a reference (Hall and Waugh, 1994), which makes a difference ...*
*NEW: ... parts of the community have established the stratospheric entry point as a reference (Hall and Plumb, 1994), which makes a difference ...*

*Comment:* **page 3, line 11: suggest "source and sink terms" for clarity**

*Reply: agreed*

*Action: "and sink" inserted.*

*Comment:* **page 4, line 2: "shallowness approximation": it will help some readers to spell out the approximation (i.e. z much smaller than $r_E$)**

*Reply: agreed*

*Action:inserted "[which] simplifies the equations using the assumption that z is much smaller than $r_E$ and which [is...]*

*Comment:* **page 7, line 28: differentiate**

Reply: agreed, thanks for spotting

Action:*corrected.*

Comment: **page 10, line 4: I'd prefer "identity matrix"**

Reply: agreed.

Action: *"matrix" inserted after "identity".*

Comment: **page 12, line 4: equal to an integer multiple ...**

Reply: agreed

Action:*"with" replaced by "to".*

Comment: **page 12, line 12: I suggest "real atmosphere"**

Reply: agreed.

Action:*"true" replaced by "real".*

Comment: **page 19, line 22/23: velocities should be v and w, subscript for $K_\phi$ (also Fig. caption of Fig. 3)**

Reply: Agreed. Thanks for spotting.

Action:*corrected.*

Comment: **page 20, line 13: "roughly consistent" should be specified/quantified more - in what way are they consistent (list a few examples, such as overall latitudinal and vertical structures etc)?**

Reply: Agreed. Specified as reported below.

Action:*We have inserted "...in a sense that the quotient of the typical circulation velocity and the distance between equator to pole gives an age estimate of the correct order of magnitude".*

Comment: **page 20, line 14/15: "shows many detailed features demanding scientific investigation in their own right": which detailed features and why to they demand investigation? I think this should be discussed in the text.**

Reply: We think that it is too early to scientifically discuss the results when the proof of concept is still an issue. However, in order to give a flavour of what can be investigated with this method, we now mention a couple of features visible in our example shown, which may be interesting research issues.

Action:*We have inserted: "(e.g., the latitude offset between the intertropical convergence zone and the stratospheric tropical pipe, or the interfacing between the stratospheric two-cell circulation and the overturning mesospheric circulation and the transition altitude between them)"*

Comment: **page 20, line 17: should be "mixing coefficients"**

Reply: agreed, thanks for spotting.

Action:*Corrected.*

Comment: **page 20, line 25: weekly**

Reply: agreed, thanks for spotting.

Action:*Corrected.*

Comment: **page 20, line 33: view _of_ the same problem**

Reply: agreed, thanks for spotting.

Action:*corrected.*

Comment: **Fig. 1: right-hand column misses a color bar; labels are hard to read - please increase font size**

Reply: agreed.

Action:*The figure has been replaced*

Comment: **Fig. 2 caption: "a detail of this" - I suggest "a zoomed-in view"**

Reply: agreed

Action:*reworded as suggested.*

**Report 2**

Comment: **General comment:**
**The paper presents a method for deducing atmospheric circulation**

(wind field) and mixing parameters from trace gas measurements by inversion of the continuity equation. In a first step, the mathematical framework is defined and explained. Second, the method is applied to idealized tracer fields and to MIPAS satellite measurements (in the "proof of concept" section 5), to show that the inversion indeed results in reliable velocities and diffusivities. Deducing information about the circulation from measurements, without involving information from models, is a great challenge in atmospheric sciences. This paper seems to contain an important contribution to reach that goal, what renders it definitely publishable and of great interest to a large readership of ACP. However, I have two major points which the authors need to assess before publication. First, the paper is not easy-to-read and the presentation quality needs improvement - otherwise I feel that the paper will fail in addressing a large readership.

Reply: see related specific replies below.

Action:*see below.*

Comment: **Second, I have some concerns about the so-called "proof of concept".**

Reply: A 'proof of concept', we think, is something much more modest than, say, a 'validation'. A proof of concept shall provide evidence that the chosen pathway is likely to lead to interesting results when pursued further. It is not meant as a full test of the functionality. The validation in terms of comparison with models etc. is a full project in its own right.

Action:*see below.*

Comment: **Major comments:**
**1) Presentation and Notation:**
**Overall, the paper is overloaded with detailed formulae, but lacking motivating and explanatory paragraphs. In their own words (P2, L33), the authors aim to avoid "that the reader does not see the forest for the trees" but, in my opinion, there are still too many trees around. For instance in section 3, there should be a clear motivation at the beginning, why the derivatives (which are calculated in the following) are needed and what the matrix notation means.**

Reply: Well, removing the trees also would mean removing part of the forest. Instead of removing equations, we prefer to better guide the reader through the equations by adding some explanatory text.

Action:
*text before Eq. 27 modified to "With these expressions, the prediction of air*

*density and volume mixing ratio can be rewritten in matrix notation for a single micro time increment. This notation simplifies the estimation of the uncertainties of the predicted atmospheric state and the inversion of the prediction equation. In matrix notation, the prediction reads"*
*added before Eq. 77: "i.e. we linearly predict the new atmospheric state for a given initial state as a function of wind and mixing ratios"*
*added after Eq. 77: "This formulation gives access to the winds and mixing ratios via inversion of* **F**.*"*

Comment: **After that the equations (15-26) could be nicely combined into one single equation-array (similarly in section 4, starting with equation (37)).**

Reply: With this we would lose all the explaining comments between the equations.

Action:*none*

Comment: **Concerning all formulae, writing X; $\chi$; $\mu$; ... for mixing ratio instead of vmr would help to increase readability.**

Reply: With all the other variable names we have detected ambiguities. $X$ (however lower case) is used as generalized atmospheric state variable (from Eq. 31 on), and $\chi$ would be confusing because $\chi^2$ is our cost function in Eq. 78. Admittedly, $\mu$ is not used in our paper but its use to designate mixing ratios is not that much established while it is often used to designate other quantities. Following Wikipedia, it designates a measure in measure theory, minimalization in computability theory and recursion theory, the mean in the normal distribution, the integrating factor in ordinary differential equations, the learning rate in artificial neural networks, the Möbius function in number theory, the population mean or expected value in probability and statistics, the coefficient of friction, the reduced mass in the two-body problem, linear density, or mass per unit length, in strings and other one-dimensional objects, permeability in electromagnetism, the magnetic dipole moment of a current-carrying coil, dynamic viscosity in fluid mechanics, the amplification factor or voltage gain of a triode vacuum tube, the electrical mobility of a charged particle, the muon, the chemical potential of a system, and many others but not mixing ratio. Thus we prefer to use the self-explaining variable name $vmr_g$

Action:*none*

Comment: **Moreover, while many steps in the calculation are written in detail (like taking derivatives), at some points I was not able to understand the derivations in detail. One such example is the matrix notation in equation (27). First, a clear motivation should be given why this matrix notation is advantageous and what it means (this is**

**the heart of the paper).**

Reply: well, as usual, matrix notation saves a lot of writing (both in the paper and in the computer code). To better motivate our matrix formalism, a paragraph on the logical flow of the operations has been inserted, where also particular advantages of the matrix notation are highlighted.

Action:*In the Section "General Concept", the logical flow of the operations is outlined. For this purpose, the following paragraph has been added: "Our concept involves the following operations. First, a general solution of the forward problem is formulated (Section 3). The forward problem is the solution of the prediction equation as a function of the initial atmospheric state for given winds and mixing coefficients. For our chosen solver, which involves the MacCormack (1969) integration scheme for the solution of the transport problem (Eqs. 5–10), the relevant dependencies of the final state on the initial state are reported in Section 3.2 (Eqs. 15-26). The formulation in matrix notation (Eqs 27–28) allows the easy handling of multiple successive timesteps (Eq. 29) and an easy estimation of the prediction error via generalized Gaussian error estimation (Eq. 30). As a next step, the dependence of the predicted state on winds and mixing coefficients is estimated for a given initial state . This is achieved by differentiation of the solution of the prediction equation with respect to winds and mixing coefficients (Eqs 34–76). These partial derivatives form the Jacobian matrix of the problem, with which the estimation of winds and mixing coefficients can be reduced to a constrained least square optimization problem where the inversely variance-weighted residual between the predicted atmospheric state and the respective measured atmospheric state is minimized. The latter step involves the generalized inverse of the Jacobian matrix (Eqs. 78–90)."*

Comment: **Second, I did not succeed in understanding the dimensionalities of the quantities involved. As the authors state, the D-matrix is build from three submatrices of dimensions $K_0 \times K_0$ ($I_K$), $K_0 \times 2K_0$ ($W_i$), and $J_0 \times L_0$ ($D_\rho; nom$). Therefore, the D-matrix has dimensions $(2K_0 + J_0) \times (3K_0 + L_0)$, which is, as far as I can see, not consistent with the vector it is acting on. Please check the dimensionalities again...**

Reply: Contrary to what we have state in our preliminary reply, the reviewer is right. Sorry for the premature author comment!

Action:*Equations have been corrected. Corrected dimensionalities are now reported.*

Comment: **... and explain clearly what equation (27) means. Equation (35) caused me similar problems with understanding. Please explain clearly where it comes from.**

Reply: All entries and the rationale behind them are explained in the list after

this equation (admittedly hard to understand with the dimensionality error in the discussion version, which has now been corrected, see above. Instead of adding more words, we have decided to visualize which matrix blocks operate on which elements of the atmospheric state vector.

Action:*A figure has been added which illustrates which parts of the D-matrix operate on which part of the input vector.*

Comment: **The appendix is, in my opinion, not necessary. It just presents a recalculation of the existing literature. I would recommend to reduce such recalculations, but to add explanations at the critical and new steps of this paper (e.g., around Eqns. 27/35). If the authors want to keep that part, it could be moved to the supplementary material.**

Reply: agreed.

Action:*As stated above, we have removed the old appendix because it was incorrect. In the new appendix we do not recalculate anything. We only report Eq. D14 of Tung (1982) which we have rewritten in our system and notation, list the definitions of involved terms, and present the interpretations of the 2D velocities and mixing coefficients inferred from this.*

*Comment:* **2) "Proof of concept": In my opinion, section 5.3 does not really present a "proof of concept", as promised by the title of section 5. The method is used to deduce velocities and diffusivities from tracer measurements, but the true underlying circulation is not known. Therefore, this case is no proof that the inversion method yields the correct result. I think, for a true "proof of concept" the circulation and diffusivities must be known before and need to be reproduced by the method. Section 5.2 points into this direction, but is exclusively descriptive. The optimal "proof of concept" would be to have a 2D-model based on equations (3-4) with idealized velocities and diffusivities, and to invert the resulting trace gas distributions. At least, the cases described in section 5.2 should be explained in more detail and related results should be shown in the paper.**

*Reply: As said above, a 'proof of concept', we think, is something much more modest than, say, a 'validation'. A proof of concept shall provide evidence that the chosen pathway is likely to lead to interesting results when pursued further and that the scheme is in principle functional. It is not meant as a full test of the functionality. The validation in terms of comparison with models etc. is a full project in its own right and is planned as future activity. Reporting and discussing all of the functionality tests performed during the development of the program would fill a book (but a boring one!). Such tests are often so trivial that the reader does not learn a lot. To give the reader a flavor of what the tests were*

*like, we have decided to report one representative example.*

*Action:We have selected a testcase as an example and discuss it.*

*Comment:***Specific comments:**
**P2, L1ff: Another source of uncertainty when deducing mean age from SF6 is related to the fact that the tropospheric increase is not strictly linear (see Garcia et al., 2011, JAS).**

*Reply: We agree in part. The non-linearity of tropospheric $SF_6$ is not an issue in itself. E.g., Stiller et al (2012) or Haenel et al. (2015) use o nonlinear tropospheric reference curve to infer the age of air from the tracer measurements. Up to that point, the inference of the age of air is unambiguous as long as the tropospheric growth is monotonical. Nonlineary becomes only a problem by mixing processes. If tropospheric growth was strictly linear, all effects implied by the asymmetric age spectrum would cancel out and no convolution with the age spectrum would be needed to obtain the correct mean age of air. In cases of nonlinear tropospheric increase, the consideration of the age spectrum becomes relevant. But, e.g. Stiller et al. (2012) or Haenel et al (2015) consider this. These authors use an iterative scheme to infer the age. As first guess, they use the age directly inferred from the SF6 mixing ratio. Then, they calculate the age spectrum for this initially guessed age, convolve the (non-linear) tropospheric time series with this age-dependent age spectrum, infer a correction, and iterate until convergence. Thus, the only remaining uncertainty in this context is the uncertainty of the age spectrum. However, we agree that not every reader might be aware of this method, and we will add a note on the non-linearity issue.*

*Action:We have changed the text as follows:*
*OLD: ...by an age spectrum, on which some ad hoc assumptions have to be made...*
*NEW: ...by an age spectrum, which has to be considered since the tropospheric growth of $SF_6$ mixing ratios is not strictly linear, and on which some ad hoc assumptions have to be made....*

*Comment:* **P2, L7: To my knowledge, in models usually the surface layer is used as a reference, not the upper edge of the TTL. Please clarify.**

*Reply: Some people insist that the upper edge of the TTL is the only adequate reference. We admit that these people are not necessarily modellers.*

*Action:We have changed the text as follows*
*OLD: ... the modelling community has established the upper edge of the tropical tropopause layer as a reference (Hall and Waugh, 1994), which makes a difference ...*
*NEW: ... parts of the community have established the stratospheric entry point*

*as a reference (Hall and Plumb, 1994), which makes a difference ...*

*Comment:* **P18, L11: How robust are the deduced velocities and diffusivities with respect to the choice of initial value for the iteration. Please give some quantitative estimate.**

*Reply: In a constrained retrieval, where the initial guess is set equal to the a priori field, there are generally two possible mechanisms for a dependence of the result on the initial guess.*

> 1. *The effect of the constraint: The solution of a constrained inversion always has a tendency to be pulled towards the a priori field, or, in our case with 1st order Tikhonov constraint, the field gradients are constrained to those of the a priori field. Our a priori constraint is chosen as zero velocity throughout. Since the resulting field gradients deviate largely from this initial guess and resemble those of the expected velocity fields, it is evident that the inversion is able to find a solution which is far away from the initial guess and is not overly constrained by the a priori assumption.*

> 2. *Non-linearity and possible secondary minima of the cost function: The observed convergence rate indicates an almost linear inverse problem; thus, no such related problems are seen. We have experienced that problematic cases regularily end up in non-converging iterations rather than converging to different results and are thus easily sorted out.*

*Our results provide the smoothest field of velocities and the smallest mixing coefficients which are still consistent with the measurements. Any change of the a priori assumptions must have an impact on the results. How large this intentional dependence is, is fully user-controllable by adjustment of the regularisation strength.*

*Action:none.*

*Comment:* **P20, L7: How can the residual be small for $SF_6$ if no chemical sink is included in the calculation? Is the sink effect absorbed in the transport terms, or is a significant sink only existing above the upper boundary for the calculation?**

*Reply: The problem with the sink in Stiller et al. (2012) is, that age-calculations are sensitive to the accumulated decomposition since the air has left the troposphere. In our case, only the decomposition during the finite time-step of the calculation can contribute,because the atmospheric state at the beginning of the time step already is depleted in the trace gases. For SF6, the major sink is indeed above the altitude range under consideration.*

*Action:none.*

*Comment:* **P20, L13: "Velocities are roughly consistent with mean ages...".** Some misinterpretations in the past arose from relating mean age simply to the stratospheric circulation. However, mean age is known to be controlled by circulation and mixing (e.g., Neu and Plumb, 1999; Garny et al., 2014; Ploeger et al., 2015). So please discuss carefully what you mean here with "consistent".

*Reply: We emphasize the attribute "roughly". This statement is not meant as a quantitative assessment but is meant to say that there is no obvious major contradiction between our results and our current knowledge on stratospheric mean age of air. We will change the text towards a more careful wording.*

*Action:We have inserted "...in a sense that the quotient of the typical circulation velocity and the distance between equator to pole gives an age estimate of the correct order of magnitude".*

Comment: **Technical corrections:**

**Equation (11): I think there should be no minus here (in the supplement there is also no minus: Eq. (34)).**

Reply: Thanks for spotting!

Action:*corrected.*

Comment: **Equation (13): Brackets missing around the argument of volume mixing ratio.**

Reply: Thanks for spotting!

Action:*corrected.*

Comment: **Equation (18): Missing point behind equation.**

Reply: Thanks for spotting!

Action:*corrected.*

Comment: **Equation (27): Are the dimensions correct - see my major comment 1.**

Reply: No! Contrary to what we have stated in the preliminary reply, dimensions are indeed incorrect.

Action:*corrected.*

Comment: **P12, L15: Point behind "numerical artefacts."**

Reply: agreed.

Action:*corrected.*

Comment: **P13, L6: K's should have an index j, like v and w.**

Reply: yes, indeed.

Action:*corrected.*

Comment: **P13, L15: vmr in italics - or better: use some symbol instead (e.g., X, see also my major comment 1).**

Reply: agreed for the italics.

Action:*corrected*

Comment: **P18, L27: "set to zero."**

Reply: agreed.

Action:*corrected.*

Comment: **P19, L13: Replace "In order to fight..." by "Due to..."**

Reply: We find the original wording more specific.

Action:

Comment: **P19, L27: "...macro timestep"**

Reply: agreed.

Action:*corrected.*

Comment: **P20, L21: I guess you mean Figure 2.**

Reply: yes indeed, thanks for spotting!

Action:*corrected.*

Comment: **P20, L23: Figure 3 has no middle column.**

Reply: I cannot find a middle column either.

Action:*corrected!*

Comment: **Figure 1: The figure, and particularly the descriptions need to be enlarged.**

Reply: Agreed

Action:*Figure as a whole, and particularly the labels have been enlarged, and the figure is now represented in the way that the rightmost colourbar is visible.*

Comment: **Figure 2: Give a color bar for the velocities in the upper/left panel. Caption: "...(upper right panel). Bottom panels show...". And write $K_\phi$ instead of $K_{phi}$.**

Reply: This is not easily possible because the length of the arrows and their colours represent a quadratic norm involving vertical and horizontal speeds of different units. No simple unit can be assigned to the length of the arrows. The colours just represent the length of the arrow and are meant simply to guide the eye. Instead we report (in the upper part of the plot) the values of horizontal and vertical windspeed that were used for the calculation of the norm. Agreed for $K_{phi}$.

Action:$K_{phi}$ *changed to* $K_\phi$.

Comment: **Figure 3: The lower/right panel is the same as the upper/right - it should show $K_z$.**

Reply: Thanks for spotting!

Action:*corrected.*

Comment: **Caption: Use $v; w$ instead of $v_\phi; \dots$ and $K_\phi$, to be consistent with the rest of the paper.**

Reply: agreed.

Action:*corrected.*

In the following we include a version of the manuscript with the differences marked. Since, however, the LaTeXdiff program available did not produce a source file which LaTeX would compile, a lot of hand editing was necessary, which naturally is prone to errors. In case of doubt the manuscript file without the marked differences should be used as reference.

[revised manuscript text omitted]

&\approx (x_{\mathrm{m};I} - x_0 - \mathbf{F}(q - q_0))^T \mathbf{S}_\mathrm{r}^{-1}\\
&\quad (x_{\mathrm{m};I} - x_0 - \mathbf{F}(q - q_0))
\end{aligned}$$

where $x_{\mathrm{m};I}$ is the measured state at the end of the macro time step and $\mathbf{S}_\mathrm{r}$ is the error covariance matrix of the residual, which is, under the assumption that prediction error and measurement errors are uncorrelated, the sum of the prediction covariance matrix and the measurement covariance matrix, both after the macro time step:

$$\mathbf{S}_\mathrm{r} = \mathbf{S}_{\mathrm{m},I} + \mathbf{S}_\mathrm{p}, \tag{79}$$

where $\mathbf{S}_\mathrm{p}$ is an $J \times J$-matrix containing those elements of $\mathbf{S}_I$ which are relevant to $x_I$. $\mathbf{S}_{\mathrm{m},I}$ is the measurement error covariance matrix of the atmospheric state after the macro time step. The minimization of the cost function gives the following estimate $\hat{q}$ of winds and mixing coefficients:

$$\hat{q} = q_0 + \left(\mathbf{F}^T\mathbf{S}_\mathrm{r}^{-1}\mathbf{F}\right)^{-1}\mathbf{F}^T\mathbf{S}_\mathrm{r}^{-1}(x_{\mathrm{m};I} - x_I) \tag{80}$$

The matrix $\mathbf{F}^T\mathbf{S}_\mathrm{r}^{-1}\mathbf{F}$ can be singular either because the related system of equations is under-determined or ill-posed due to nearly linearly dependent equations. Singularity is

fought by adding the following constraint term to the cost function of Eq. (78):

$$\begin{aligned}
\chi_{\mathrm{con}}^2 &= (q - q_a)^T \mathbf{R}(q - q_a) \tag{81}\\
\chi^2 &= \chi_1^2 + \chi_{\mathrm{con}}^2, \tag{82}
\end{aligned}$$

where $q_a$ is some prior assumption on velocities and mixing coefficients. $\mathbf{R}$ is a $J \times J$ regularization matrix of which the choice is discussed below. From this, the constrained estimate of velocities and mixing coefficients can be inferred:

$$\hat{q} = q_a + \left(\mathbf{F}^T\mathbf{S}_\mathrm{r}^{-1}\mathbf{F} + \mathbf{R}\right)^{-1}\mathbf{F}^T\mathbf{S}_\mathrm{r}^{-1}(x_{\mathrm{m};I} - x_I) \tag{83}$$

An equivalent formulation, which is more efficient if the dimension of $q$ is larger than that of $x$ (underdetermined problem), but which requires a non-singular regularization matrix  and does not give easy access to diagnostics (see below), is (Rodgers, 2000):

$$\hat{q} = q_a + \mathbf{R}^{-1}\mathbf{F}^T\left(\mathbf{F}\mathbf{R}^{-1}\mathbf{F}^T + \mathbf{S}_{\mathrm{r},I}\right)^{-1}(x_{\mathrm{m};I} - x_I). \tag{84}$$

The covariance matrix characterizing the uncertainty of estimated winds and mixing coefficients is

$$\mathbf{S}_q = \left(\mathbf{F}^T\mathbf{S}_\mathrm{r}^{-1}\mathbf{F} + \mathbf{R}\right)^{-1}\mathbf{F}^T\mathbf{S}_\mathrm{r}^{-1}\mathbf{F}\left(\mathbf{F}^T\mathbf{S}_\mathrm{r}^{-1}\mathbf{F} + \mathbf{R}\right)^{-1}, \tag{85}$$

and the estimated winds and mixing coefficients are related to the true ones as

$$\mathbf{A} = \frac{\partial\hat{q}}{\partial q} = \left(\mathbf{F}^T\mathbf{S}_\mathrm{r}^{-1}\mathbf{F} + \mathbf{R}\right)^{-1}\mathbf{F}^T\mathbf{S}_\mathrm{r}^{-1}\mathbf{F}, \tag{86}$$

which is unity in the case of unconstrained estimation of $q$. In the case of Newtonian iteration, Eqs. (85-86) are evaluated using the Jacobian $\mathbf{F}$ valid at the solution.

Due to the concentration-dependence of the source function and the $q$-depecdence of $\mathbf{F}$, Eq. (29) is valid only in linear approximation. This is helped by putting the inversion in the context of a Newtonian iteration (see, e.g., Rodgers (2000, p. 85). Eq. (80) becomes

$$\hat{q}_{it+1} = q_{it} + \left(\mathbf{F}_{it}^T\mathbf{S}_\mathrm{r}^{-1}\mathbf{F}_{it}\right)^{-1}\mathbf{F}_{it}^T\mathbf{S}_\mathrm{r}^{-1}(x_{\mathrm{m};I} - x_{I,it}), \tag{87}$$

where $it$ is the iteration index. Equation (83) becomes

$$\begin{aligned}
\hat{q}_{it+1} &= q_{it} + \left(\mathbf{F}_{it}^T\mathbf{S}_\mathrm{r}^{-1}\mathbf{F}_{it} + \mathbf{R}\right)^{-1} \tag{88}\\
&\quad \left(\mathbf{F}_{it}^T\mathbf{S}_\mathrm{r}^{-1}(x_{\mathrm{m};I} - x_{I,it}) - \mathbf{R}(q_{it} - q_a)\right)
\end{aligned}$$

or alternatively

$$\begin{aligned}
\hat{q}_{it+1} &= q_a + \left(\mathbf{F}_{it}^T\mathbf{S}_\mathrm{r}^{-1}\mathbf{F}_{it} + \mathbf{R}\right)^{-1} \tag{89}\\
&\quad \mathbf{F}_{it}^T\mathbf{S}_\mathrm{r}^{-1}\left(x_{\mathrm{m};I} - x_{I,it} + \mathbf{F}_{it}(q_{it} - q_a)\right),
\end{aligned}$$

and Eq. (84) becomes

$$\begin{aligned}
\hat{q}_{it+1} &= q_a + \mathbf{R}^{-1}\mathbf{F}_{it}^T\left(\mathbf{F}_{it}\mathbf{R}^{-1}\mathbf{F}_{it}^T + \mathbf{S}_{\mathrm{r},I}\right)^{-1} \tag{90}\\
&\quad \left(x_{\mathrm{m};I} - x_{I,it} + \mathbf{F}_{it}(q_{it} - q_a)\right).
\end{aligned}$$

With $\boldsymbol{q}_a = \mathbf{0}$ and diagonal $\mathbf{R} = \gamma\mathbf{I}$ we get the smallest possible velocities and mixing coefficients still consistent with the measurement, where tuning parameter $\gamma$ will be set depending on how large fit residuals the user still considers to be 'consistent'. With $\mathbf{R}$ being diagonally blockwise composed of squared and scaled first order finite differences operators and $\boldsymbol{q}_a = \mathbf{0}$, smooth fields of wind vectors and mixing coefficients can be enforced. Setting $\boldsymbol{q}_a$ the result of the previous macro time step  corresponds to sequential data assimilation. In this application $\mathbf{R}$  is set to the reciprocal uncertainty of $\boldsymbol{q}_a$ plus some margin for allowance of variability of velocity and mixing coefficients in time. And finally, if prior knowledge is formed by independent measurements and their reciprocal uncertainties as constraint matrix, or within the debatable framework of Bayesian statistics, estimates $\hat{\boldsymbol{q}}$ would even be the most probable estimate of velocities and mixing ratios.

**5 Proof of concept**

**5.1 Prediction of the atmospheric state**

In a first step we test the predictive power of the formalism defined by Eqs. (3–29). Since the formalism itself is deductive and starts from a well established theoretical concept, the purpose of the test is solely to verify that the implementation of the formalism is correct and that involved numerical approximations are adequate. As a consequence of the Bonini paradox (c.f. Bonini (1963) and Starbuck (1975)), a model is the harder to understand the more complex it is. While the predictive power of a model usually increases with complexity, this does not necessarily hold for its explanatory power. Thus we have decided to test our model on the basis of very simple test cases, where major failure of the model is immediately obvious. Four test cases have been chosen, each dedicated to one kinematic variable ($v$, $w$, $K_\phi$ and $K_z$), while the other three were set to zero.

[revised manuscript text omitted]

---

## Referee Report (RR1)

**Referee comment on "Direct Inversion of Circulation and Mixing from Tracer Measurements: I. Method" by T. von Clarmann and U. Grabowski**

Although the discussion paper included already an impressive analysis, the revised version has further improved significantly, in particular regarding the presentation style (e.g., the new Fig. 1, illustrating the D-matrix), and I recommend publication in ACP as is. A few typoes I detected while reading are listed below (line numbers refer to the version acp-2016-322-manuscript-version3.pdf):

L169: there is a blank before the '.'
L171: ...(Eqs. 34-76)...
L894: phi → \phi
L895: ...shown IN Fig. 2b...
L897: ...veloCity...
L897: ...Eq. (91)...
L898: ...(Eq. 28)...
L913: The residual between … (no OF)
L916: (Fig. 2d) → point missing
Figure 2: units of the color bar of Fig. 2A should be given – at least in the caption.
L977: ...to Eq. (85) ...